# Dissociable neural correlates of uncertainty underlie different exploration strategies

Momchil S. Tomov[1,2 ✉], Van Q. Truong [2], Rohan A. Hundia[2] & Samuel J. Gershman[2]

Most real-world decisions involve a delicate balance between exploring unfamiliar alternatives and committing to the best known option. Previous work has shown that humans rely on different forms of uncertainty to negotiate this "explore-exploit" trade-off, yet the neural basis of the underlying computations remains unclear. Using fMRI ($n = 31$), we find that relative uncertainty is represented in right rostrolateral prefrontal cortex and drives directed exploration, while total uncertainty is represented in right dorsolateral prefrontal cortex and drives random exploration. The decision value signal combining relative and total uncertainty to compute choice is reflected in motor cortex activity. The variance of this signal scales with total uncertainty, consistent with a sampling mechanism for random exploration. Overall, these results are consistent with a hybrid computational architecture in which different uncertainty computations are performed separately and then combined by downstream decision circuits to compute choice.

[1] Program in Neuroscience, Harvard Medical School, Boston, MA 02115, USA. [2] Department of Psychology and Center for Brain Science, Harvard University, Cambridge, MA 02138, USA. ✉email: mtomov@g.harvard.edu

For every decision that we make, we have to choose between the best option that we know so far (exploitation), or a less familiar option that could be even better (exploration). This "explore-exploit" dilemma appears at all levels of decision making, ranging from the mundane (Do I go to my favorite restaurant, or try a new one?) to the momentous (Do I marry my current partner, or try a new one?). Despite its ubiquity in everyday life, little is known about how the brain handles the exploration-exploitation trade-off. Intuitively, either extreme is undesirable: an agent that solely exploits will never adapt to changes in the environment (or even learn which options are good in the first place), while an agent that solely explores will never reap the fruits of that exploration. Yet striking the perfect balance is computationally intractable beyond the simplest examples, and hence humans and animals must adopt various heuristics[1–3].

Earlier research suggested that people choose options in proportion to their expected values[4,5], a strategy known as softmax exploration that is closely related to other psychological phenomena such as probability matching (for a detailed review, see Schulz and Gershman[6]). Later studies showed that people explore in a more sophisticated manner, using uncertainty to guide their choices towards more promising options[7,8]. These strategies are more adaptive in nonstationary environments and come in two distinct flavors: directed and random exploration strategies.

Directed exploration strategies direct the agent's choices toward uncertain options, which is equivalent to adding an uncertainty bonus to their subjective values. For example, for your next meal, you might forego your favorite restaurant for a new one that just opened down the street, and you might even go there several times until you are certain it is no better than your favorite. Thus while softmax exploration is sensitive to the relative value of each option, preferring options with higher payoffs, directed exploration is additionally sensitive to the relative uncertainty of each option, preferring options with more uncertainty as they hold greater potential for gain. Directed exploration is closely related to the phenomenon of risk-seeking[9,10] and has strong empirical support[11–13].

Previous work[8,14,15] has shown that directed exploration in humans is well captured by the upper confidence bound (UCB) algorithm[16], in which the uncertainty bonus is the one-sided confidence interval of the expected value:

$$a_t = \arg\max_k [Q_t(k) + U_t(k)], \qquad (1)$$

where $a_t$ is the action chosen at time $t$, $Q_t(k)$ is the expected reward of action $k$ at time $t$, and $U_t(k)$ is the upper confidence bound of the reward that plays the role of an uncertainty bonus. In a Bayesian variant of UCB[17], $Q_t(k)$ corresponds to the posterior mean and $U_t(k)$ is proportional to the posterior standard deviation $\sigma_t(k)$. Returning to the restaurant example, even if both restaurants have the same expected value ($Q_t(\text{new}) = Q_t(\text{old})$), UCB would initially prefer the new one since it has greater uncertainty ($U_t(\text{new}) > U_t(\text{old})$).

Random exploration strategies introduce randomness into choice behavior, causing the agent to sometimes explore less favorable options. For example, when you move to a new neighborhood, you might initially pick restaurants at random until you learn which ones are good. While earlier studies favored value-based random exploration strategies such as softmax exploration, later work[8,14] has shown that random exploration in people is additionally sensitive to the total uncertainty of the available options, increasing choice stochasticity when option values are more uncertain. This can cause choice variability to track payoff variability, a phenomenon sometimes referred to as the payoff variability effect[18–20].

One prominent instantiation of random exploration in reinforcement learning is Thompson sampling[21], which samples values randomly from the posterior value distribution of each action and then chooses greedily with respect to the sampled values:

$$\widetilde{Q}_t(k) \sim p(Q_t(k)) \qquad (2)$$

$$a_t = \arg\max_k \widetilde{Q}_t(k), \qquad (3)$$

where $p(\cdot)$ is the posterior value distribution and $\widetilde{Q}_t(k)$ is the sampled value for arm $k$ at time $t$. Returning to the neighborhood example, the familiar restaurants in your old neighborhood have narrow value distributions around their expected values (low total uncertainty). This will cause Thompson sampling to consistently draw samples $\widetilde{Q}_t(k)$ that are close to their expected values, which will often result in choosing the same restaurant, namely the one with the highest expected value. In contrast, the unfamiliar restaurants in the new neighborhood have wide value distributions (high total uncertainty), which will result in significant variation in the Thompson samples $\widetilde{Q}_t(k)$ and a corresponding variation in the chosen restaurant.

Directed and random exploration strategies confer separate ecological advantages, which has led researchers in reinforcement learning to develop algorithms that use a hybrid of UCB and Thompson sampling[22,23]. Correspondingly, recent evidence suggests that people also employ a combination of directed and random exploration[7,8]. A study by Gershman[14] used a two-armed bandit task to show that human choices are consistent with a particular hybrid of UCB and Thompson sampling. Furthermore, the study showed that different uncertainty computations underlie each exploration strategy, with the relative uncertainty between the two options driving directed exploration, and the total uncertainty of the two options driving random exploration. This led us to hypothesize the existence of dissociable neural implementations of both strategies in the brain. At least three lines of evidence support this claim. First, dopamine genes with anatomically distinct expression profiles are differentially associated with directed and random exploration[15]. Second, transcranial magnetic stimulation of right rostrolateral prefrontal cortex (RLPFC) affects directed, but not random, exploration[24]. Third, directed and random exploration have different developmental trajectories[25].

In the present study, we use functional MRI to probe the neural underpinnings of the uncertainty computations that influence directed and random exploration. Subjects perform a two-armed bandit task in which each arm was either "safe", meaning it delivers the same reward during the whole block, or "risky", meaning it delivers variable rewards. This allows us to separate the effects of relative and total uncertainty and examine how their neural correlates influence directed and random exploration, in accordance with the theoretical principles outlined above. We find that relative uncertainty is reflected in right RLPFC, and total uncertainty is reflected in right dorsolateral prefrontal cortex (DLPFC), replicating findings reported by Badre et al.[26]. The neural signal in right RLPFC predicts cross-trial variability in directed but not random exploration, whereas the neural signal in right DLPFC predicts cross-trial variability in random but not directed exploration. We also find that the linear combination of relative and total uncertainty with value is reflected in motor cortex, suggesting that these upstream estimates are integrated by motor circuits in order to compute the categorical decision. By linking activity in those regions with human choices via a hybrid UCB/Thompson sampling model, our work provides new insight into the distinct uncertainty computations performed by the brain and their role in guiding behavior.

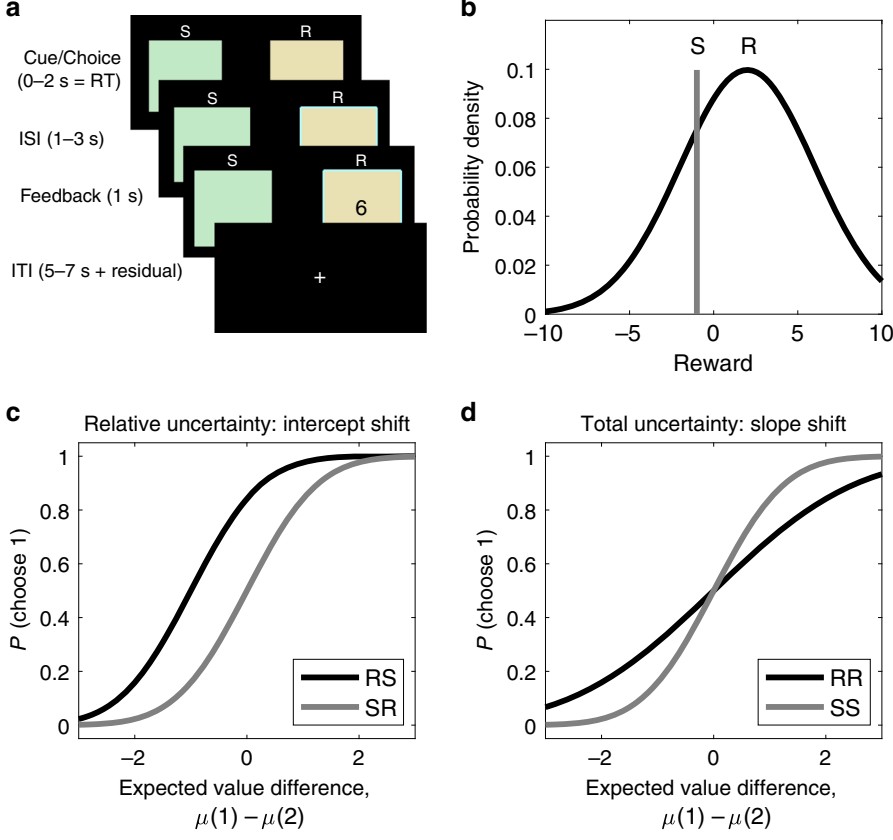

**Fig. 1 Experimental Design and Predictions. a** Trial structure. Subjects choose between two options, each labeled as either safe (S) or risky (R). After they make a choice, they receive feedback in the form of points. Option labels remain constant during the block. **b** Reward structure. Risky options deliver rewards drawn from a Gaussian distribution whose mean remains constant during the block. Safe options deliver the same reward during the block. The means of both options are resampled from the zero-mean Gaussian at the start of each block. **c** Directed exploration (UCB) predicts a bias towards the uncertain option, which shifts the choice probability function in the opposite directions for RS and SR trials. **d** Random exploration (Thompson sampling) predicts more randomness when uncertainty is high, which reduces the slope of the choice probability function for RR compared to SS trials.

## Results

**Relative and total uncertainty guide directed and random exploration.** We scanned 31 human subjects (17 female, ages 18–35) using functional MRI while they performed a two-armed bandit task in which subjects are informed about the riskiness of each option[14]. On each trial, subjects saw the labels of the two options, each of which could be either "safe" (S) or "risky" (R; Fig. 1a). A safe option always delivered the same reward for the duration of the block, while a risky option delivered Gaussian-distributed rewards around a mean that remained fixed for the duration of the block (Fig. 1b). We denote the trial types by the pair of option labels (e.g., on "RS" trials, option 1 is risky and option 2 is safe). Trial types and average rewards remained fixed within blocks and varied randomly across blocks. Subjects were explicitly informed of the statistics of the task and performed four practice blocks before entering the scanner.

Importantly, this task design allowed us to independently measure the effect of different types of uncertainty on subject choices. Directed and random exploration predict different effects across different block conditions, which can be illustrated by considering the probability of choosing option 1, $P$(choose 1), as a function of the expected value difference for the given block, $\mu(1) - \mu(2)$ (the choice function; Fig. 1c, d).

RS and SR trials manipulate relative uncertainty (greater for the option 1 on RS trials and greater for option 2 on SR trials) while controlling for total uncertainty (identical across RS and SR trials). A strategy that is insensitive to relative uncertainty such as softmax exploration would be indifferent between the two options

$(P(\text{choose } 1) = 0.5)$ when they have equal values $(\mu(1) - \mu(2) = 0)$. In contrast, UCB predicts a bias towards the risky option, preferring option 1 on RS trials and option 2 on SR trials, even when the expected value difference might dictate otherwise. This would manifest as an opposite intercept shift in the choice probability function of RS and SR trials, such that $P(\text{choose } 1) > 0.5$ when $\mu(1) - \mu(2) = 0$ on RS trials and $P(\text{choose } 1) < 0.5$ when $\mu(1) - \mu(2) = 0$ on SR trials (Fig. 1c).

In contrast, RR and SS trials manipulate total uncertainty (high on RR trials and low on SS trials) while controlling for relative uncertainty (identical across RR and SS trials). A strategy that is insensitive to total uncertainty such as UCB would predict the same choice function for both trial types. In contrast, Thompson sampling predicts more stochastic choices when there is more uncertainty, resulting in more random choices ($P$(choose 1) closer to 0.5) even when the relative expected value strongly favors one option ($\mu(1) - \mu(2)$ far from 0). This would manifest as a shallower slope of the choice probability function of RR compared to SS trials (Fig. 1d).

Overall, subjects identified the better option in each block (i.e., the option $k$ with the greater expected reward $\mu(k)$) relatively quickly, with average performance plateauing by the middle of each block (Supplementary Fig. 1). Importantly, in accordance with the theory, we found that manipulating relative uncertainty (RS vs. SR) shifted the intercept of the choice probability function (Fig. 2a, Supplementary Fig. 2): the intercept for RS trials was significantly greater than the intercept for SR trials ($F(1, 9711) = 21.0$, $p = 0.000005$). Moreover, the intercept for RS trials was significantly

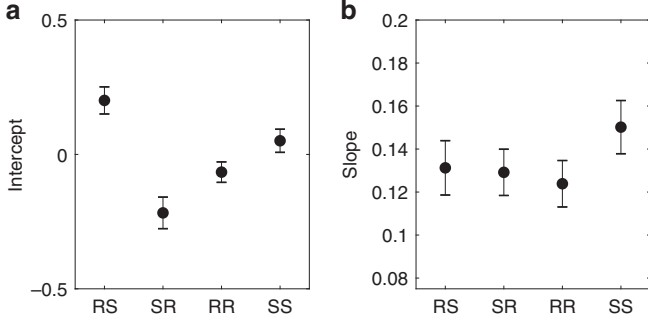

**Fig. 2 Probit regression results. a** Intercept and **b** slope of choice probability function fit for each condition using maximum likelihood estimation. Error bars are cross-subject standard errors.

greater than 0 ($F(1, 9711) = 10.8$, $p = 0.001$), while the intercept for SR trials was significantly less than 0 ($F(1, 9711) = 17.9$, $p = 0.00002$). There was only a small effect of total uncertainty on the intercept (RR vs. SS, $F(1, 9711) = 4.1$, $p = 0.04$). This indicates that, regardless of relative expected value, subjects showed a bias towards the risky option, consistent with UCB.

Conversely, manipulating total uncertainty (RR vs. SS) altered the slope of the choice probability function (Fig. 2b, Supplementary Fig. 2): the slope for RR trials is smaller than the slope for SS trials ($F(1, 9711) = 3.4$, $p = 0.07$). While the effect is small due to the small sample size, it is in the right direction and is consistent with previous replications of this experiment[14,15]. There was no effect of relative uncertainty (RS vs. SR) on the slope ($F(1, 9711) = 0.06$, $p = 0.8$). This indicates that when both options were risky, subjects were less sensitive to their relative reward advantage, consistent with Thompson sampling.

To examine how relative and total uncertainty influence directed and random exploration on a trial-by-trial basis, we modeled subject choices using a probit regression model[14]:

$$P(a_t = 1 | \mathbf{w}) = \Phi(w_1 V_t + w_2 \mathrm{RU}_t + w_3 V_t / \mathrm{TU}_t), \quad (4)$$

where $\Phi(\cdot)$ is the standard Gaussian cumulative distribution function and the regressors are the following model-derived trial-by-trial posterior estimates:

- Value difference, $V_t = Q_t(1) - Q_t(2)$.
- Relative uncertainty, $\mathrm{RU}_t = \sigma_t(1) - \sigma_t(2)$.
- Total uncertainty, $\mathrm{TU}_t = \sqrt{\sigma_t^2(1) + \sigma_t^2(2)}$.

here $Q_t(k)$ corresponds to the posterior expected value of option $k$ (Eq. (6)) and $\sigma_t(k)$ is the posterior standard deviation around that expectation (Eq. (7)), proportional to the uncertainty bonus in UCB. Note that these are trial-by-trial estimates based on the posterior quantities computed by the ideal observer model.

Gershman[8] showed that, despite its apparent simplicity, this is not a reduced form model but rather the exact analytical form of the most parsimonious hybrid of UCB and Thompson sampling that reduces to pure UCB when $w_3 = 0$, to pure Thompson sampling when $w_2 = 0$, and to pure softmax exploration when $w_2 = w_3 = 0$. Thus the hybrid model balances exploitation (governed by $w_1$) with directed ($w_2$) and random ($w_3$) exploration simultaneously for each choice, without the need to dynamically select one strategy over the other (whether and how the brain might perform this meta-decision is beyond the scope of our present work). If subjects use both UCB and Thompson sampling, the model predicts that all three regressors will have a significant effect on choices ($w_1 > 0$, $w_2 > 0$, $w_3 > 0$).

Correspondingly, the maximum likelihood estimates of all three fixed effects coefficients were significantly greater than zero:

$w_1 = 0.166 \pm 0.016$ ($t(9716) = 10.34$, $p < 10^{-20}$; mean ± s.e.m., two-tailed $t$-test), $w_2 = 0.175 \pm 0.021$ ($t(9716) = 8.17$, $p < 10^{-15}$), and $w_3 = 0.005 \pm 0.001$ ($t(9716) = 4.47$, $p < 10^{-5}$). Model comparisons revealed that the UCB/Thompson hybrid model fits subject choices better than UCB or Thompson sampling alone, which in turn fit choices better than softmax alone (Supplementary Table 1). Bayesian model comparison strongly favored the hybrid model over alternative models (protected exceedance probability = 1[27]).

Furthermore, running these models generatively with the corresponding fitted parameters on the same bandits as the subjects revealed significant differences in model performance (Supplementary Fig. 3, $F(3, 1236) = 291.58$, $p < 10^{-20}$, one-way ANOVA). The UCB/Thompson hybrid outperformed UCB and Thompson sampling alone (UCB vs. hybrid, $p < 10^{-8}$; Thompson vs. hybrid, $p < 10^{-8}$, pairwise multiple comparison tests), which in turn outperformed softmax exploration (softmax vs. UCB, $p < 10^{-5}$; softmax vs. Thompson, $p < 10^{-8}$). Similar results replicated across a range of coefficients (Supplementary Fig. 4), signifying the distinct and complementary ecological advantages of UCB and Thompson sampling. Thus relying on both UCB ($w_2 > 0$) and Thompson sampling ($w_3 > 0$) should yield better overall performance. In line with this prediction, we found better performance among subjects whose choices are more sensitive to $\mathrm{RU}_t$ (greater $w_2$), consistent with greater reliance on UCB (Supplementary Fig. 5B, $r(29) = 0.47$, $p = 0.008$, Pearson correlation). Similarly, we found better performance among subjects whose choices are more sensitive to $V_t / \mathrm{TU}_t$ (greater $w_3$), consistent with greater reliance on Thompson sampling (Supplementary Fig. 5C, $r(29) = 0.53$, $p = 0.002$). Finaly, note that even though optimal exploration is intractable in general, the hybrid model computes choices in constant time by simply computing Eq. (4). Taken together, these results replicate and expand upon previous findings[14], highlighting the superiority of the UCB/Thompson hybrid as a descriptive as well as normative model of uncertainty-guided exploration. Thus humans do and ought to employ both directed and random exploration, driven by relative and total uncertainty, respectively.

**Neural correlates of relative and total uncertainty**. Next, we asked whether relative and total uncertainty are represented in distinct anatomical loci. We performed an unbiased whole-brain univariate analysis using a general linear model (GLM 1) with model-derived trial-by-trial posterior estimates of the quantities used in computing the decision (Eq (4)): absolute relative uncertainty ($|\mathrm{RU}_t|$), total uncertainty ($\mathrm{TU}_t$), absolute value difference ($|V_t|$), and absolute value difference scaled by total uncertainty ($|V_t|/\mathrm{TU}_t$) as non-orthogonalized impulse regressors at trial onset (see Methods section). We report whole-brain $t$-maps after thresholding single voxels at $p < 0.001$ (uncorrected) and applying cluster family wise error (FWE) correction at significance level $\alpha = 0.05$.

For relative uncertainty, we found a large negative bilateral occipital cluster that extended dorsally into inferior parietal cortex and primary motor cortex, and ventrally into inferior temporal cortex (Supplementary Fig. 7A, Supplementary Table 3). For total uncertainty, we found bilateral positive clusters in the inferior parietal lobe, DLPFC, anterior insula, inferior temporal lobe, midcingulate cortex, superior posterior thalamus, and premotor cortex (Supplementary Fig.7B, Supplementary Table 4). We did not find any clusters for $|V_t|$ or $|V_t|/\mathrm{TU}_t$.

Based on previous studies[24,26], we expected to find a positive cluster for relative uncertainty in right RLPFC. While we did observe such a cluster in the uncorrected contrast (Fig. 3a), it did not survive FWE correction. We pursued this hypothesis further using a priori ROIs from Badre et al.[26], who reported a positive

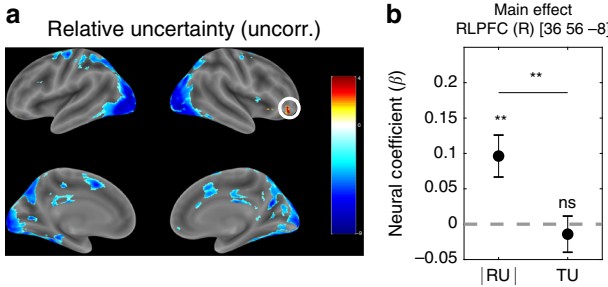

**Fig. 3 Right RLPFC tracks relative but not total uncertainty. a** Whole-brain $|RU_t|$ contrast from GLM 1. Single voxels were thresholded at $p < 0.001$. Multiple comparisons correction was not applied (corrected version is shown in Supplementary Fig. 7A). The color scale represents $t$-values across subjects. The circled ROI in right RLPFC (MNI [36 56 − 8]) from Badre et al.[26] was used in the subsequent confirmatory analysis (10-mm sphere around the peak voxel). **b** Neural regression coefficients (betas) from GLM 1 for the parametric modulators $|RU_t|$ ($\beta_{|RU|}$) and $TU_t$ ($\beta_{TU}$) at trial onset, averaged across voxels in the ROI. Error bars are cross-subject standard errors. Comparisons were made using Student's $t$-tests. $^{**}p < 0.01$, ns: not significant. Source data are provided as a Source Data file.

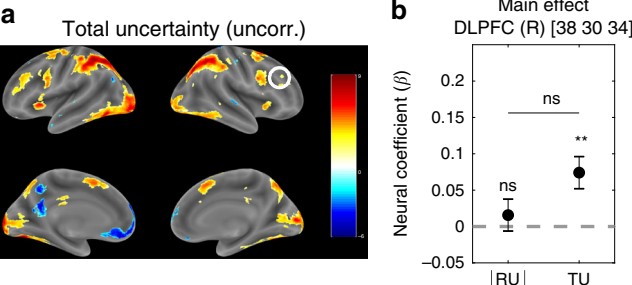

**Fig. 4 Right DLPFC tracks total but not relative uncertainty. a** Whole-brain $TU_t$ contrast from GLM 1. Single voxels were thresholded at $p < 0.001$. Multiple comparisons correction was not applied (corrected version is shown in Supplementary Fig. 7B). The color scale represents $t$-values across subjects. The circled ROI in right DLPFC (MNI [38 30 34]) from Badre et al.[26] was used in the subsequent confirmatory analysis (10 mm sphere around the peak voxel). **b** Neural regression coefficients (betas) from GLM 1 for the parametric modulators $|RU_t|$ ($\beta_{|RU|}$) and $TU_t$ ($\beta_{TU}$) at trial onset. Error bars are cross-subject standard errors. Comparisons were made using Student's $t$-tests. $^{**}p < 0.01$, ns: not significant. Source data are provided as a Source Data file.

effect of relative uncertainty in right RLPFC (MNI [36 56 − 8]) and of total uncertainty in right DLPFC (MNI [38 30 34]). In accordance with their results, in right RLPFC we found a significant effect of relative uncertainty (Fig. 3b; $t(30) = 3.24$, $p = 0.003$, two-tailed t-test) but not of total uncertainty ($t(30) = −0.55$, $p = 0.58$). The significance of this difference was confirmed by the contrast between the two regressors ($t(30) = 2.96$, $p = 0.006$, paired t-test). Conversely, in right DLPFC there was a significant effect of total uncertainty (Fig. 4b; $t(30) = 3.36$, $p = 0.002$) but not of relative uncertainty ($t(30) = 0.71$, $p = 0.48$), although the contrast between the two did not reach significance ($t(30) = 1.74$, $p = 0.09$). These results replicate Badre et al.[26]'s findings and suggest that relative and total uncertainty are represented in right RLPFC and right DLPFC, respectively.

**Subjective estimates of relative and total uncertainty predict choices.** If right RLPFC and right DLPFC encode relative and total uncertainty, respectively, then we should be able to use their activations to decode trial-by-trial subjective estimates of $RU_t$ and

$TU_t$. In particular, on any given trial, a subject's estimate of relative and total uncertainty might differ from the ideal observer estimates $RU_t$ and $TU_t$ stipulated by the hybrid model (Eq. (4)). This could occur for a number of reasons, such as neural noise, inattention, or a suboptimal learning rate. Importantly, any deviation from the ideal observer estimates would result in a corresponding deviation of the subject's choices from the model predictions. Therefore if we augment the hybrid model to include the neurally decoded subjective estimates of relative and total uncertainty (denoted by $\widehat{RU_t}$ and $\widehat{TU_t}$, respectively), then we should arrive at more accurate predictions of subject choices (see Methods section).

Indeed, this is what we found. Including the decoded trial-by-trial $\widehat{RU_t}$ (Eq. (12)) from right RLPFC (MNI [36 56 −8]) significantly improved predictions of subject choices (Table 1; BICs: 6407 vs. 6410). Importantly, decoding trial-by-trial $\widehat{TU_t}$ from right RLPFC and augmenting the model with $V_t/\widehat{TU_t}$ (Eq. (13)) did not improve choice predictions (BICs: 6421 vs. 6410).

Similarly, augmenting the hybrid model with $V_t/\widehat{TU_t}$ (Eq. (13)) when $\widehat{TU_t}$ was decoded from right DLPFC (MNI [38 30 34]) significantly improved predictions of subject choices (Table 1; BICs: 6359 vs. 6410). Conversely, augmenting the model with $\widehat{RU_t}$ (Eq. (12)) decoded from right DLPFC did not improve choice predictions (BICs: 6419 vs. 6410). Together, these results show that variability in the neural representations of uncertainty in the corresponding regions predicts choices, consistent with the idea that those representations are used in a downstream decision computation.

We additionally augmented the model with both $\widehat{RU_t}$ from right RLPFC and $V_t/\widehat{TU_t}$, with $\widehat{TU_t}$ from right DLPFC (Eq. (14), Table 1). This improved choice predictions beyond the improvement of including $\widehat{RU_t}$ alone (BICs: 6359 vs. 6406). It also resulted in better choice fits than including $V_t/\widehat{TU_t}$ alone (Eq. (13)), which is reflected in the lower AIC (6273 vs. 6287) and deviance (6249 vs. 6266), even though the more stringent BIC criterion is comparable (6359 vs. 6359). This suggests that the two uncertainty computations provide complementary yet not entirely independent contributions to choices.

**Neural correlates of downstream decision value computation.** We next sought to identify the downstream decision circuits that combine the relative and total uncertainty estimates to compute choice. Following the rationale of the UCB/Thompson hybrid model, we assume that the most parsimonious way to compute decisions is to linearly combine the uncertainty estimates with the value estimate, as in Eq. (4). We therefore employed a similar GLM to GLM 1 (GLM 2, Supplementary Table 2) with model-derived trial-by-trial estimates of the decision value (DV) as the only parametric modulator at trial onset. We quantified decision value as the linear combination of the terms in Eq. (4), weighted by the corresponding subject-specific random effects coefficients **w** from the probit regression:

$$DV_t = w_1 V_t + w_2 RU_t + w_3 V_t / TU_t. \quad (5)$$

As before, we took the absolute decision value $|DV_t|$ for purposes of identifiability. As previously, we thresholded single voxels at $p < 0.001$ (uncorrected) and applied cluster FWE correction at significance level $\alpha = 0.05$. This revealed a single negative cluster in left primary motor cortex (peak MNI [−38 −8 62], Fig. 5a, Supplementary Table 5).

We defined an ROI as a 10-mm sphere around the peak voxel, which we refer to as left M1 in subsequent confirmatory analyses. Note that this activation is not simply reflecting motor responses,

**Table 1 Model comparison with neurally decoded regressors.**

| Model | Regressors | AIC | BIC | LL | Deviance |
|---|---|---|---|---|---|
| **Baseline** | | | | | |
| UCB/Thompson hybrid with intercept (Eq. (4)) | $1 + V + \text{RU} + V/\text{TU}$ | 6352.75 | 6410.00 | − 3168.37 | 6336.75 |
| **$\widehat{\text{RU}}$ and $\widehat{\text{TU}}$ from right RLPFC** | | | | | |
| Baseline augmented with $\widehat{\text{RU}}$ (Eq. (12)) | $1 + V + \text{RU} + V/\text{TU} + \widehat{\text{RU}}$ | 6334.97 | 6406.54 | − 3157.48 | 6314.97 |
| Baseline augmented with $\widehat{\text{TU}}$ (Eq. (13)) | $1 + V + \text{RU} + V/\text{TU} + V/\widehat{\text{TU}}$ | 6350.00 | 6421.57 | − 3165.00 | 6330.00 |
| **$\widehat{\text{RU}}$ and $\widehat{\text{TU}}$ from right DLPFC** | | | | | |
| Baseline augmented with $\widehat{\text{RU}}$ (Eq. (12)) | $1 + V + \text{RU} + V/\text{TU} + \widehat{\text{RU}}$ | 6347.28 | 6418.85 | − 3163.64 | 6327.28 |
| Baseline augmented with $\widehat{\text{TU}}$ (Eq. (13)) | $1 + V + \text{RU} + V/\text{TU} + V/\widehat{\text{TU}}$ | 6286.96 | 6358.53 | − 3133.48 | 6266.96 |
| **$\widehat{\text{RU}}$ from right RLPFC and $\widehat{\text{TU}}$ from right DLPFC** | | | | | |
| Baseline augmented with $\widehat{\text{RU}}$ and $\widehat{\text{TU}}$ (Eq. (14)) | $1 + V + \text{RU} + V/\text{TU} + \widehat{\text{RU}} + V/\widehat{\text{TU}}$ | 6273.14 | 6359.03 | − 3124.57 | 6249.14 |
| **$\widehat{\text{DV}}$ from left M1** | | | | | |
| Baseline augmented with $\widehat{\text{DV}}$ (Eq. (15)) | $1 + V + \text{RU} + V/\text{TU} + \widehat{\text{DV}}$ | 6336.27 | 6407.84 | − 3158.13 | 6316.27 |

Model fits after augmenting the UCB/Thompson hybrid (Eq. (4)) with estimates of relative uncertainty, total uncertainty, and decision value, decoded from brain activity. Lower AIC, BIC, and deviance indicate better fit.
AIC: Akaike information criterion, BIC: Bayesian information criterion, LL: maximized log likelihood.

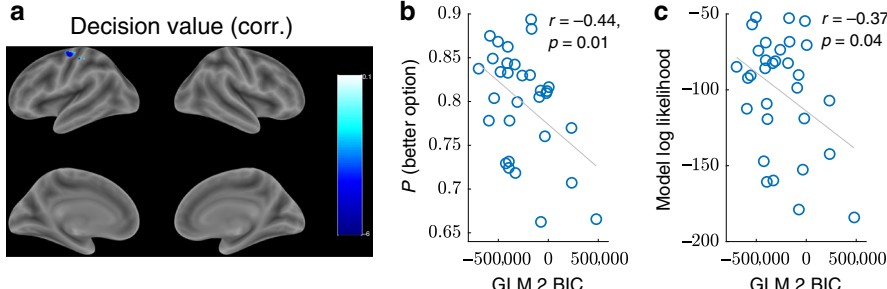

**Fig. 5 Primary motor cortex tracks decision value. a** Whole-brain $|\text{DV}_t|$ contrast from GLM 2. Single voxels were thresholded at $p < 0.001$ and cluster FWE correction was applied at significance level $\alpha = 0.05$. The ROI in left primary motor cortex (left M1; peak MNI [ − 38  − 8 62]) was used in the subsequent confirmatory analysis (10 mm sphere around the peak voxel). **b** Cross-subject Pearson correlation between the BIC in left M1, quantifying the extent to which neural activity in that region is captured by GLM 2 (lower BIC indicates better fit), and average performance. **c** Cross-subject Pearson correlation between the BIC in left M1 and model log likelihood, quantifying the extent to which the subject's choices are consistent with the UCB/Thompson hybrid model. $r$: Pearson correlation coefficient.

since those are captured by a chosen action regressor (Supplementary Table 2). Another possible confound is reaction time (RT). When controlling for RT, motor cortex activations become unrelated to $|\text{DV}_t|$ (GLM 2A in Supplementary Information). This could be explained by a sequential sampling implementation of our model (see Discussion), according to which RT would depend strongly on DV. Consistent with this interpretation, model comparison revealed that left M1 activity is best explained by a combination of DV and RT, rather than DV or RT alone (see Supplementary Information).

**Subjective estimate of decision value predicts within-subject and cross-subject choice variability.** If left M1 encodes the decision value, then we should be able to use its activation to decode trial-by-trial subjective estimates of $\text{DV}_t$, similarly to how we were able to extract subjective estimates of $\text{RU}_t$ and $\text{TU}_t$. In particular, on any given trial, a subject's estimate of the decision value might differ from the linear combination of the ideal observer estimates (Eq. (5)). Importantly, any such deviations would result in corresponding deviations from the model-predicted choices. Following the same logic as before, we augmented the hybrid model to include a linearly decoded trial-by-trial estimate of the decision value (denoted by $\widehat{\text{DV}}$) from left M1 (Eq. (15)). This improved predictions of subject choices (Table 1,

BICs: 6408 vs. 6410), consistent with the idea that this region computes the linear combination of relative and total uncertainty with value, which in turn is used to compute choice.

In order to further validate the ROI, we performed a cross-subject correlation between the extent to which GLM 2 captures neural activity in left M1 (quantified by the BIC; see Methods section) and subject performance (quantified by the proportion of trials on which the subject chose the better option). We reasoned that some subjects will perform computations that are more similar to our model than other subjects. If our model is a plausible approximation of the underlying computations, then it should better capture neural activity in the decision value ROI for those subjects, resulting in lower BICs. Furthermore, those subjects should also exhibit better performance, in line with the normative principles of our model (Supplementary Figs. 3, 4, and 5; Supplementary Table 1). This prediction was substantiated: we found a significant negative correlation between BIC and performance (Fig. 5b, $r(29) = -0.44$, $p = 0.01$, Pearson correlation), indicating that subjects whose brain activity matches the model also tend to perform better. We found a similar correlation between BIC and model log likelihood (Fig. 5c, $r(29) = -0.37$, $p = 0.04$), which quantifies how well the subject's behavior is captured by the UCB/Thompson hybrid model (Eq. (4)). Together, these results build upon our previous findings and suggest that left M1 combines the subjective estimate of relative

and total uncertainty from right RLPFC and right DLPFC, respectively, with the subjective value estimate, in order to compute choice.

**Variability in the decision value signal scales with total uncertainty**. The lack of any main effect for $|V_t|/TU_t$ in GLM 1 could be explained by a mechanistic account according to which, instead of directly implementing our closed-form probit model (Eq. (4)), the brain is drawing and comparing samples from the posterior value distributions. This corresponds exactly to Thompson sampling and would produce the exact same behavior as the analytical model. However, it makes different neural predictions, namely that: (1) there would be no explicit coding of $V_t/TU_t$, and (2) the variance of the decision value would scale with (squared) total uncertainty. The latter is true because the variance of the Thompson sample for arm $k$ on trial $t$ is $\sigma_t^2(k)$, and hence the variance of the sample difference is $\sigma_t^2(1) + \sigma_t^2(2) = TU_t^2$. Thus while we cannot infer the drawn samples on any particular trial, we can check whether the unexplained variance around the mean decision value signal in left M1 is correlated with $TU_t^2$.

To test this hypothesis, we correlated the residual variance of the GLM 2 fits in the decision value ROI (Fig. 5a; left M1, peak MNI $[-38\ -8\ 62]$) with $TU_t^2$. We found a positive correlation ($t(30) = 2.06$, $p = 0.05$, two-tailed $t$-test across subjects of the within-subject Fisher z-transformed Pearson correlation coefficients), consistent with the idea that total uncertainty affects choices via a sampling mechanism that is implemented in motor cortex.

## Discussion

Balancing exploration and exploitation lies at the heart of decision making, and understanding the neural circuitry that underlies different forms of exploration is central to understanding how the brain makes choices in the real world. Here we show that human choices are consistent with a particular hybrid of directed and random exploration strategies, driven respectively by the relative and total uncertainty of the options. This dissociation between the two uncertainty computations predicted by the model was reified as an anatomical dissociation between their neural correlates. Our GLM results confirm the previously identified role of right RLPFC and right DLPFC in encoding relative and total uncertainty, respectively[26]. Crucially, our work further elaborates the functional role of those regions by providing a normative account of how both uncertainty estimates are used by the brain to make choices, with relative uncertainty driving directed exploration and total uncertainty driving random exploration. This account was validated by our decoding analysis and decision value GLM, which suggest that the two uncertainty estimates are combined with the value estimate in downstream motor cortex, which ultimately performs the categorical decision computation.

While our study replicates the results reported by Badre et al.[26], it goes beyond their work in several important ways. First, our task design explicitly manipulates uncertainty – the main quantity of interest – across the different task conditions, whereas the task design in Badre et al.[26] is focused on manipulating expected value. Second, relative and total uncertainty are manipulated independently in our task design: relative uncertainty differs across RS and SR trials, while total uncertainty remains fixed, on average; the converse holds for SS and RR trials. Orthogonalizing relative and total uncertainty in this way allows us to directly assess their differential contribution to choices (Supplementary Fig. 2). Third, the exploration strategies employed by our model are rooted in normative principles developed in the machine learning literature[16,21], with theoretical performance guarantees

which were confirmed by our simulations (Supplementary Figs. 3 and 4). In particular, the separate contributions of relative and total uncertainty to choices are derived directly from UCB and Thompson sampling, implementing directed and random exploration, respectively. Fourth, this allows us to link relative and total uncertainty and their neural correlates directly to subject behavior and interpret the results in light of the corresponding exploration strategies.

Previous studies have also found a signature of exploration in RLPFC, also referred to as frontopolar cortex[4,26,28,29], however, with the exception of Badre et al.[26], these studies did not disentangle different exploration strategies or examine their relation to uncertainty. More pertinent to our study, Zajkowski, Kossut, and Wilson[24] reported that inhibiting right RLPFC reduces directed but not random exploration. This is consistent with our finding that activity in right RLPFC tracks the subjective estimate of relative uncertainty which underlies the directed exploration component of choices in our model. Disrupting activity in right RLPFC can thus be understood as introducing noise into or reducing the subjective estimate of relative uncertainty, resulting in choices that are less consistent with directed exploration.

One important contribution of our work is to elucidate the role of the total uncertainty signal from right DLPFC in decision making. Previous studies have shown that DLPFC is sensitive to uncertainty[30] and that perturbing DLPFC can affect decision-making under uncertainty[31–33]. Most closely related to our study, Knoch et al.[31] showed that suppressing right DLPFC (but not left DLPFC) with repetitive transcranial magnetic stimulation leads to risk-seeking behavior, resulting in more choices of the suboptimal "risky" option over the better "safe" option. Conversely, Fecteau et al.[33] showed that stimulating right DLPFC with transcranial direct current stimulation reduces risk-seeking behavior. One way to interpret these findings in light of our framework is that, similarly to the Zajkowski, Kossut, and Wilson[24] study, suppressing right DLPFC reduces random exploration, which diminishes sensitivity to the value difference (Eq. (4), third term) while allowing directed exploration to dominate choices (Eq. (4), second term), leading to apparently risk-seeking behavior. Stimulating right DLPFC would then have the opposite effect, increasing random exploration and thereby increasing sensitivity to the value difference between the two options, leading to an increased preference for the safe option.

Our definition of uncertainty as the posterior standard deviation of the mean (sometimes referred to as estimation uncertainty, parameter uncertainty, or ambiguity) is different from the expected uncertainty due to the unpredictability of the reward from the risky option on any particular trial (sometimes referred to as irreducible uncertainty or risk; [34]). These two forms of uncertainty are generally related, and in particular in our study, estimation uncertainty is higher, on average, for risky arms due to the variability of their rewards, which makes it difficult to estimate the mean exactly. However, they are traditionally associated with opposite behaviors: risk-aversion predicts that people would prefer the safe over the risky option[35], while uncertainty-guided exploration predicts a preference for the risky option, all else equal (Fig. 1c). While we did not seek to explicitly disentangle risk from estimation uncertainty in our study, our behavioral (Fig. 2a and Supplementary Fig. 2) and neural (Fig. 3) results are consistent with the latter interpretation.

Our finding that decision value is reflected in motor cortex might seem somewhat at odds with previous neuroimaging studies of value coding, which is often localized to ventromedial prefrontal cortex[vmPFC;36], orbitofrontal cortex[37], or the intraparietal sulcus[38]. However, most of these studies consider the values of the available options ($Q_t$) or the difference between them ($V_t$), without taking into account the uncertainty of those

quantities. This suggests that the values encoded in those regions are divorced from any uncertainty-related information, which would render them insufficient to drive uncertainty-guided exploratory behavior on their own. Uncertainty would have to be computed elsewhere and then integrated with these value signals by downstream decision circuits closer to motor output. Our results do not contradict these studies (in fact, we observe traces of value coding in vmPFC in our data as well; see Supplementary Fig. 9C and Supplementary Information) but instead point to the possibility that the value signal is computed separately and combined with the uncertainty signals from RLPFC and DLPFC downstream by motor cortex, which ultimately computes choice.

One mechanism by which this could occur is suggested by sequential sampling models, which posit that the decision value $DV_t$ drives a noisy accumulator to a decision bound, at which point a decision is made[19]. This is consistent with Gershman's[14] analysis of reaction time patterns on the same task as ours. It is also consistent with studies reporting neural signatures of evidence accumulation during perceptual as well as value-based judgments in human motor cortex[39–43]. It is worth noting that for our right-handed subjects, left motor cortex is the final cortical area implementing the motor choice. One potential avenue for future studies would be to investigate whether the decision value area will shift if subjects respond using a different modality, such as their left hand, or using eye movements. This would be consistent with previous studies that have identified effector-specific value coding in human cortex[38].

One prediction following from the sequential sampling interpretation is that motor cortex should be more active for more challenging choices (i.e. when $DV_t$ is close to zero), since the evidence accumulation process would take longer to reach the decision threshold. Indeed, this is consistent with our result, and reconciles the apparently perplexing negative direction of the effect: $|DV_t|$ can be understood as reflecting decision confidence, since a large $|DV_t|$ indicates that one option is significantly preferable to the other, making it highly likely that it would be chosen, whereas a small $|DV_t|$ indicates that the two options are comparable, making choices more stochastic (Eq. (4) and (5)). Since we found that motor cortex is negatively correlated with $|DV_t|$, this means that it is negatively correlated with decision confidence, or equivalently, that it is positively correlated with decision uncertainty. In other words, motor cortex is more active for more challenging choices, as predicted by the sequential sampling framework.

Another puzzling aspect of our results that merits further investigation is the lack of any signal corresponding to $V_t/TU_t$. This suggests that the division might be performed by circuits downstream from right DLPFC, such as motor cortex. Alternatively, it could be that, true to Thompson sampling, the brain is generating samples from the posterior value distributions and comparing them to make decisions. In that case, what we are seeing in motor cortex could be the average of those samples, consistent with the analytical form of the UCB/Thompson hybrid (Eq. (4)) which is derived precisely by averaging over all possible samples[8]. A sampling mechanism could thus explain both the negative sign of the $|DV_t|$ effect in motor cortex, as well as the absence of $V_t/TU_t$ in the BOLD signal. Such a sampling mechanism also predicts that the variance of the decision value signal should scale with (squared) total uncertainty, which is precisely what we found. Overall, our data suggest that random exploration might be implemented by a sampling mechanism which directly enters the drawn samples into the decision value computation in motor cortex.

The neural and behavioral dissociation between relative and total uncertainty found in our study points to potential avenues for future research that could establish a double dissociation between the corresponding brain regions. Temporarily disrupting activity in right RLPFC should affect directed exploration (reducing $w_2$ in Eq. (4)), while leaving random exploration intact (not changing $w_3$ in Eq. (4)). Conversely, disrupting right DLPFC should affect random but not directed exploration. This would expand upon the RLPFC results of Zajkowski, Kossut, and Wilson[24] by establishing a causal role for both regions in the corresponding uncertainty computations and exploration strategies.

In summary, we show that humans tackle the exploration-exploitation trade-off using a combination of directed and random exploration strategies driven by neurally dissociable uncertainty computations. Relative uncertainty was correlated with activity in right RLPFC and influenced directed exploration, while total uncertainty was correlated with activity in right DLPFC and influenced random exploration. Subjective trial-by-trial estimates decoded from both regions predicted subject responding, while motor cortex reflected the combined uncertainty and value signals necessary to compute choice. Our results are thus consistent with a hybrid computational architecture in which relative and total uncertainty are computed separately in right RLPFC and right DLPFC, respectively, and then integrated with value in motor cortex to ultimately perform the categorical decision computation via a sampling mechanism.

## Methods

**Subjects**. We recruited 31 subjects (17 female) from the Cambridge community. All subjects were healthy, ages 18–35, right-handed, with normal or corrected vision, and no neuropsychiatric pre-conditions. Subjects were paid $50.00 for their participation plus a bonus based on their performance. The bonus was the number of points from a random trial paid in dollars (negative points were rounded up to 1). All subjects received written consent and the study was approved by the Harvard Institutional Review Board.

**Experimental design and statistical analysis**. We used the two-armed bandit task described in Gershman[14]. On each block, subjects played a new pair of bandits for 10 trials. Each subject played 32 blocks, with 4 blocks in each scanner run (for a total of 8 runs per subject). On each trial, subjects chose an arm and received reward feedback (points delivered by the chosen arm). Subjects were told to pick the better arm in each trial. To incentivize good performance, subjects were told that at the end of the experiment, a trial will be drawn randomly and they will receive the number of points in dollars, with negative rewards rounded up to 1. While eliminating the possibility of losses may appear to have altered the incentive structure of the task, subjects nevertheless preferred the better option across all task conditions (Supplementary Fig. 1A), in accordance with previous replications of the experiment[8,14].

On each block, the mean reward $\mu(k)$ for each arm $k$ was drawn randomly from a Gaussian with mean 0 and variance $\tau_0^2(k) = 100$. Arms on each block were designated as "risky" (R) or "safe" (S), with all four block conditions (RS, SR, RR, and SS) counterbalanced and randomly shuffled within each run (i.e., each run had one of each block condition in a random order). A safe arm delivered the same reward $\mu(S)$ on each trial during the block. A risky arm delivered rewards sampled randomly from a Gaussian with mean $\mu(R)$ and variance $\tau^2(R) = 16$. The type of each arm was indicated to subjects by the letter R or S above the corresponding box. In order to make it easier for subjects to distinguish between the arms within and across blocks, each box was filled with a random color that remained constant throughout the blocks and changed between blocks. The color was not informative of rewards.

Subjects were given the following written instructions.

In this task, you have a choice between two slot machines, represented by colored boxes. When you choose one of the slot machines, you will win or lose points. One slot machine is always better than the other, but choosing the same slot machine will not always give you the same points. Your goal is to choose the slot machine that you think will give you the most points. Sometimes the machines are "safe" (always delivering the same points), and sometimes the machines are "risky" (delivering variable points). Before you make a choice, you will get information about each machine: "S" indicates SAFE, "R" indicates RISKY. The "safe" or "risky" status does not tell you how rewarding a machine is. A risky machine could deliver more or less points on average than a safe machine. You cannot predict how good a machine is simply based on whether it is considered safe or risky. Some boxes will deliver negative points. In those situations, you should select the

one that is least negative. In the MRI scanner, you will play 32 games, each with a different pair of slot machines. Each game will consist of 10 trials. Before we begin the actual experiment, you will play a few practice games. Choose the left slot machine by pressing with your index finger and the right slot machine by pressing with your middle finger. You will have 2 seconds to make a choice. To encourage you to do your best, at the end of the MRI experiment, a random trial will be chosen and you will be paid the number of points you won on that trial in dollars. You want to do as best as possible every time you make a choice!

The event sequence within a trial is shown in Fig. 1a. At trial onset, subjects saw two boxes representing the two arms and chose one of them by pressing with their index finger or their middle finger on a response button box. The chosen arm was highlighted and, after a random inter-stimulus interval (ISI), they received reward feedback as the number of points they earned from that arm. No feedback was provided for the unchosen arm. Feedback remained on the screen for 1 s, followed by a random inter-trial interval (ITI) and the next trial. A white fixation cross was displayed during the ITI. If subjects failed to respond within 2 s, they were not given reward feedback and after the ISI, they directly entered the ITI with a red fixation cross. Otherwise, the residual difference between 2 s and their reaction time was added to the following ITI. Each block was preceded by a 6-s inter-block-interval, during which subjects saw a sign, "New game is starting," for 3 s, followed by a 3-s fixation cross. A 10-s fixation cross was added to the beginning and end of each run to allow for scanner stabilization and hemodynamic lag, respectively. ISIs and ITIs were pregenerated by drawing uniformly from the ranges 1–3 s and 5–7 s, respectively. Additionally, ISIs and ITIs in each run were uniformly scaled such that the total run duration is exactly 484 s, which is the expected run length assuming an average ISI (2 s) and an average ITI (6 s). This accounted for small deviations from the expected run duration and allowed us to acquire 242 whole-brain volumes during each run (TR = 2 s). The experiment was implemented using the PsychoPy toolbox[44].

**Belief updating model.** Following Gershman[14], we assumed subjects approximate an ideal Bayesian observer that tracks the expected value and uncertainty for each arm. Since rewards in our task are Gaussian-distributed, these correspond to the posterior mean $Q_t(k)$ and variance $\sigma_t^2(k)$ of each arm $k$, which can be updated recursively on each trial $t$ using the Kalman filtering equations:

$$Q_{t+1}(a_t) = Q_t(a_t) + \alpha_t[r_t - Q_t(a_t)] \tag{6}$$

$$\sigma_{t+1}^2(a_t) = \sigma_t^2(a_t) - \alpha_t \sigma_t^2(a_t), \tag{7}$$

where $a_t$ is the chosen arm, $r_t$ is the received reward, and the learning rate $\alpha_t$ is given by:

$$\alpha_t = \frac{\sigma_t^2(a_t)}{\sigma_t^2(a_t) + \tau^2(a_t)}. \tag{8}$$

We initialized the values with the prior means, $Q_t(k) = 0$ for all $k$, and variances with the prior variances, $\sigma_1^2(k) = \tau_0^2(k)$. Subjects were informed of those priors and performed 4 practice blocks (40 trials) before entering the scanner to familiarize themselves with the task structure. Kalman filtering is the Bayes-optimal algorithm for updating the values and uncertainties given the task structure and has been previously shown to account well for human choices in bandit tasks[4,8,12,45]. In order to prevent degeneracy of the Kalman update for safe arms, we used $\tau^2(S) = 0.00001$ instead of zero, which is equivalent to assuming a negligible amount of noise even for safe arms. Notice that in this case, the learning rate is $\alpha_t \approx 1$ and as soon as the safe arm $k$ is sampled, the posterior mean and variance are updated to $Q_{t+1}(k) \approx \mu(k)$ and $\sigma_t^2(k) \approx 0$, respectively.

**Choice probability analysis.** We fit the coefficients **w** of the hybrid model (Eq. (4)) using mixed-effects maximum likelihood estimation (fitglme in MATLAB, with *FitMethod* =Laplace, CovariancePattern = diagonal, and EBMethod = TrustRegion2D) using all non-timeout trials (i.e., all trials on which the subject made a choice within 2 s of trial onset). In Wilkinson notation[46], the model specification was: Choice ∼ V + RU + VoverTU+ (V + RU + VoverTU | SubjectID).

We confirmed the parameter recovery capabilities of our approach by running the hybrid model generatively on the same bandits as the subjects[47]. We drew the weights in each simulation according to $\mathbf{w} \sim \mathcal{N}(0, 10 \times \mathbf{I})$ and repeated the process 1000 times. We found a strong correlation between the generated and the recovered weights (Supplementary Fig. 6A; $r > 0.99$, $p < 10^{-8}$ for all weights) and no correlation between the recovered weights (Supplementary Fig. 6B; $r < 0.03$, $p > 0.3$ for all pairs), thus validating our approach. We fit the lesioned models in Supplementary Table 1 in the same way.

To generate Supplementary Fig. 4, we similarly ran the model generatively, but this time using a grid of 16 evenly spaced values between 0 and 1 for each coefficient. For every setting of the coefficients **w**, we computed performance as the proportion of times the better option was chosen, averaged across simulated subjects, averaged across 10 separate iterations. We preferred this metric over total reward as it yields more comparable results across different bandit pairs. To

generate Supplementary Fig. 3, we similarly ran the model generatively, but this time using the fitted coefficients.

In addition to the hybrid model, we also fit a model of choices as a function of experimental condition to obtain the slope and intercept of the choice probability function:

$$P(a_t = 1|\mathbf{w}) = \Phi\left(\sum_j w_1^j \pi_{tj} + w_2^j \pi_{tj} V_t\right), \tag{9}$$

where $j$ is the experimental condition (RS, SR, RR, or SS), and $\pi_{tj} = 1$ if trial $t$ is assigned to condition $j$, and 0 otherwise. In Wilkinson notation, the model specification was: Choice ∼ condition + condition : V + (condition + condition : V | SubjectID). We plotted the $w_1$ terms as the intercepts and the $w_2$ terms as the slopes.

For Bayesian model comparison, we fit **w** separately for each subject using fixed effects maximum likelihood estimation (Choice ∼ V + RU + VoverTU) in order to obtain a separate BIC for each subject. We approximated the log model evidence for each subject as $-0.5*\text{BIC}$ and used it to compute the protected exceedance probability for each model, which is the probability that the model is most prevalent in the population[27].

**fMRI data acquisition.** We followed the same protocol as described previously[48]. Scanning was carried out on a 3T Siemens Magnetom Prisma MRI scanner with the vendor 32-channel head coil (Siemens Healthcare, Erlangen, Germany) at the Harvard University Center for Brain Science Neuroimaging. A T1-weighted high-resolution multi-echo magnetization-prepared rapid-acquisition gradient echo (ME-MPRAGE) anatomical scan[49] of the whole brain was acquired for each subject prior to any functional scanning (176 sagittal slices, voxel size = 1.0 × 1.0 × 1.0 mm, TR = 2530 ms, TE = 1.69–7.27 ms, TI = 1100 ms, flip angle = 7°, FOV = 256 mm). Functional images were acquired using a T2*-weighted echo-planar imaging (EPI) pulse sequence that employed multiband RF pulses and Simultaneous Multi-Slice (SMS) acquisition[50–52]. In total, eight functional runs were collected for each subject, with each run corresponding to four task blocks, one in each condition (84 interleaved axial-oblique slices per whole-brain volume, voxel size = 1.5 × 1.5 × 1.5 mm, TR = 2000 ms, TE = 30 ms, flip angle = 80°, in-plane acceleration (GRAPPA) factor = 2, multi-band acceleration factor = 3, FOV = 204 mm). The initial 5 TRs (10 s) were discarded as the scanner stabilized. Functional slices were oriented to a 25° tilt towards coronal from AC-PC alignment. The SMS-EPI acquisitions used the CMRR-MB pulse sequence from the University of Minnesota.

All 31 scanned subjects were included in the analysis. We excluded runs with excessive motion (>2 mm translational motion or >2° rotational motion). Four subjects had a single excluded run and two additional subjects had two excluded runs.

**fMRI preprocessing.** As in our previous work[48], functional images were pre-processed and analyzed using SPM12 (Wellcome Department of Imaging Neuroscience, London, UK). Each functional scan was realigned to correct for small movements between scans, producing an aligned set of images and a mean image for each subject. The high-resolution T1-weighted ME-MPRAGE images were then co-registered to the mean realigned images and the gray matter was segmented out and normalized to the gray matter of a standard Montreal Neurological Institute (MNI) reference brain. The functional images were then normalized to the MNI template (resampled voxel size 2 mm isotropic), spatially smoothed with a 8-mm full-width at half-maximum (FWHM) Gaussian kernel, high-pass filtered at 1/ 128 Hz, and corrected for temporal autocorrelations using a first-order autoregressive model.

**Univariate analysis.** Our hypothesis was that different brain regions perform the two kinds of uncertainty computations (relative and total uncertainty), which in turn drive the two corresponding exploration strategies (directed exploration, operationalized as UCB, and random exploration, operationalized as Thompson sampling). We therefore defined a general linear model (GLM 1, Supplementary Table 2) with model-based trial-by-trial posterior estimates of absolute relative uncertainty, $|RU_t|$, total uncertainty, $TU_t$, absolute value difference, $|V_t|$, and absolute value difference scaled by total uncertainty, $|V_t|/TU_t$, as parametric modulators for an impulse regressor at trial onset (trial_onset). All quantities were the same model-derived ideal observer trial by trial estimates which we used to model choices (Eq. (4)). For ease of notation, we sometimes refer to those parametric modulators as RU, TU, V, and V/TU, respectively. Following[26], we used $|RU_t|$ instead of $RU_t$ to account for our arbitrary choice of arm 1 and arm 2 (note that $TU_t$ is always positive). We used the absolute value difference $|V_t|$ for the same reason.

The trial_onset regressor was only included on trials on which the subject responded within 2 s of trial onset (i.e., non-timeout trials). We included a separate regressor at trial onset for trials on which the subject timed out (i.e., failed to respond within 2 s of trial onset) that was not parametrically modulated (trial_onset_timeout), since failure to respond could be indicative of failure to perform the necessary uncertainty computations. In order to control for any motor-related activity that might be captured by those regressors due to the

hemodynamic lag, we also included a separate trial onset regressor on trials on which the subject chose arm 1 (trial_onset_chose_1). Finally, to account for response-related activity and feedback-related activity, we included regressors at reaction time (button_press) and feedback onset (feedback_onset), respectively. All regressors were impulse regressors (duration = 0 s) convolved with the canonical hemodynamic response function (HRF). The parametric modulators were not orthogonalized[53]. As is standard in SPM, there was a separate version of each regressor for every scanner run. Additionally, there were six motion regressors and an intercept regressor.

For group-level whole-brain analyses, we performed $t$-contrasts with single voxels thresholded at $p < 0.001$ and cluster family-wise error (FWE) correction applied at $\alpha = 0.05$, reported in Supplementary Fig. 7 and Supplementary Tables 3 and 4. Uncorrected contrasts are shown in Figs. 3 and 4. We labeled clusters based on peak voxel labels from the deterministic Automated Anatomical Labeling (AAL2) atlas[54,55]. For clusters whose peak voxels were not labeled successfully by AAL2, we consulted the SPM Anatomy Toolbox[56] and the CMA Harvard-Oxford atlas[57]. We report up to 3 peaks per cluster, with a minimum peak separation of 20 voxels. All voxel coordinates are reported in Montreal Neurological Institute (MNI) space.

Since the positive cluster for $|RU_t|$ in right RLPFC did not survive FWE correction, we resorted to using a priori ROIs from a study by Badre et al.[26] for our subsequent analysis. Even though in their study subjects performed the different task and the authors used a different model, we believe the underlying uncertainty computations are equivalent to those in our study and hence likely to involve the same neural circuits. We defined the ROIs as spheres of radius 10-mm around the peak voxels for the corresponding contrasts reported by Badre et al.[26]: right RLPFC for relative uncertainty (MNI [36 56 −8]) and right DLPFC for total uncertainty (MNI [38 30 34]).

To compute the main effect in a given ROI, we averaged the neural coefficients (betas) within a sphere of radius 10-mm centered at the peak voxel of the ROI for each subject, and then performed a two-tailed $t$-test against 0 across subjects. To compute a contrast in a given ROI, we performed a paired two-tailed $t$-test across subjects between the betas for one regressor (e.g. $|RU_t|$) and the betas for the other regressor (e.g. $|TU_t|$), again averaged within a 10-mm-radius sphere.

We used the same methods for the decision value GLM (GLM 2, Supplementary Table 2) as with GLM 1.

**Decoding.** If the brain regions reported by Badre et al.[26] encode subjective trial-by-trial estimates of relative and total uncertainty, as our GLM 1 results suggest, and if those estimates dictate choices, as our UCB/Thompson hybrid model predicts, then we should be able to read out those subjective estimates and use them to improve the model predictions of subject choices. This can be achieved by "inverting" the GLM and solving for $|RU_t|$ and $TU_t$ based on the neural data $y$, the beta coefficients $\beta$, and the design matrix $X$. Using ridge regression to prevent overfitting, the subjective estimate for relative uncertainty for a given voxel on trial $t$ can be computed as:

$$|\widehat{RU}_t| = \left(y_t - \sum_{i:X_{t,i}\neq|RU|} X_{t,i}\beta_i\right)\beta_{|RU|}/(\beta_{|RU|}^2 + \lambda) \quad (10)$$

where $y_t$ is the neural signal on trial $t$, $X_{t,i}$ is the value of regressor $i$ on trial $t$, $\beta_i$ is the corresponding beta coefficient computed by SPM, $\beta_{|RU|}$ is the beta coefficient for $|RU|$, and $\lambda$ is the ridge regularization constant (voxel indices were omitted to keep the notation uncluttered). The sum is taken over all regressors $i$ other than $|RU|$. To account for the hemodynamic lag, we approximated the neural activity at time $t$ as the raw BOLD signal at time $t + 5$ s, which corresponds to the peak of the canonical HRF in SPM (spm_hrf). Since the beta coefficients were already fit by SPM, we could not perform cross-validation to choose $\lambda$ and so we arbitrarily set $\lambda = 1$.

To obtain a signed estimate for relative uncertainty, we flipped the sign based on the model-based $RU_t$:

$$\widehat{RU}_t = \begin{cases} |\widehat{RU}_t| & \text{if } RU_t >= 0 \\ -|\widehat{RU}_t| & \text{if } RU_t < 0 \end{cases} \quad (11)$$

Note that our goal is to test whether we can improve our choice predictions, given that we already know the model-based $RU_t$, so this does not introduce bias into the analysis. Finally, we averaged $\widehat{RU}_t$ across all voxels within the given ROI (a 10-mm sphere around the peak voxel) to obtain a single trial-by-trial subjective estimate $\widehat{RU}_t$ for that ROI. We used the same method to obtain a trial-by-trial subjective estimate of total uncertainty, $\widehat{TU}_t$.

To check if the neurally-derived $\widehat{RU}_t$ predicts choices, we augmented the probit regression model of choice in Eq. (4) to:

$$P(a_t = 1|\mathbf{w}) = \Phi(w_0 + w_1 V_t + w_2 RU_t + w_3 V_t/TU_t + w_4 \widehat{RU}_t). \quad (12)$$

In Wilkinson notation[46], the model specification was: `Choice ~ 1 + V + RU + VoverTU + decodedRU + (1 + V + RU + VoverTU + decodedRU | SubjectID)`.

Notice that we additionally included an intercept term $w_0$. While this departs from the proper analytical form of the UCB/Thompson hybrid (Eq. (4)), we

noticed that including an intercept term alone is sufficient to improve choice predictions (data not shown), indicating that some subjects had a bias for one arm over the other. We therefore chose to include an intercept to guard against false positives, which could occur, for example, if we decode low-frequency noise which adopts the role of a de facto intercept.

We then fit the model in the same way as the probit regression model in Eq. (4), using mixed effects maximum-likelihood estimation (fitglme), with the difference that we omitted trials from runs that were excluded from the fMRI analysis. For baseline comparison, we also re-fitted the original model (Eq. (4)) with an intercept and without the excluded runs (hence the difference between the UCB/Thompson hybrid fits in Table 1 and Supplementary Table 1).

Similarly, we defined an augmented model for $\widehat{TU}_t$:

$$P(a_t = 1|\mathbf{w}) = \Phi(w_0 + w_1 V_t + w_2 RU_t + w_3 V_t/TU_t + w_4 V_t/\widehat{TU}_t), \quad (13)$$

Note that this analysis cannot be circular since it evaluates the ROIs based on behavior, which was not used to define GLM 1[58]. In particular, all regressors and parametric modulators in GLM 1 were defined purely based on model-derived ideal observer quantities; we only take into account subjects' choices when fitting the weights $\mathbf{w}$ (Eq. (4)), which were not used to define the GLM 1. Furthermore, since all model-derived regressors are also included in all augmented models of choice (Eqs. (12), (13), and (14)), any additional choice information contributed by the neurally decoded regressors is necessarily above and beyond what was already included in GLM 1.

Finally, we entered both subjective estimates from the corresponding ROIs into the same augmented model:

$$P(a_t = 1|\mathbf{w}) = \Phi(w_0 + w_1 V_t + w_2 RU_t + w_3 V_t/TU_t + w_4 \widehat{RU}_t + w_5 V_t/\widehat{TU}_t), \quad (14)$$

We similarly constructed an augmented model with the decoded decision value, $\widehat{DV}_t$, from GLM 2, after adjusting the sign similarly to Eq. (11):

$$P(a_t = 1|\mathbf{w}) = \Phi(w_0 + w_1 V_t + w_2 RU_t + w_3 V_t/TU_t + w_4 \widehat{DV}_t), \quad (15)$$

**Residual variance analysis.** To show that the variance of the decision value signal scales with total uncertainty, we extracted the residuals of the GLM 2 fits from the $|DV_t|$ ROI (Fig. 5; left M1, peak MNI [−38 −8 62]), averaged within a 10-mm sphere around the peak voxel. As with the decoding analysis, we accounted for the hemodynamic lag by taking the residuals 5 s after trial onset to correspond to the residual activations on the given trial. We then performed a Pearson correlation between the square of the residuals (the residual variance) and $TU_t^2$ across trials for each subject. Finally, to aggregate across subjects, we Fisher z-transformed the resulting correlation coefficients and performed a two-tailed one sample $t$-test against zero.

**Reporting summary.** Further information on research design is available in the Nature Research Reporting Summary linked to this article.

## Data availability
All behavioral data are available at https://github.com/tomov/Exploration-fMRI-Task. The raw fMRI data is available upon request. The source data underlying Figs. 3b and 4b are provided as a Source Data file. A reporting summary for this Article is available as a Supplementary Information file.

## Code availability
All analyses were conducted in MATLAB using SPM 12 and our custom fMRI analysis pipeline built on top of it, which is available at https://github.com/sjgershm/ccnl-fmri. All analysis code is available at https://github.com/tomov/Exploration-Data-Analysis.

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

## Acknowledgements

This research was supported by the Office of Naval Research Science of Autonomy program (N00014-17-1-2984), the National Institutes of Health (award number 1R01MH109177), and the Toyota Corporation. This work involved the use of instrumentation supported by the NIH Shared Instrumentation Grant Program award number S10OD020039. This work also received funding from the Irene and Eric Simon Brain Research Foundation. We acknowledge the University of Minnesota Center for Magnetic Resonance Research for use of the multiband-EPI pulse sequences. We are grateful to Wouter Kool and Eric Schulz for giving us feedback on the manuscript.

## Author contributions

S.G. and M.T. conceptualized and designed the study. V.T. and M.T. implemented the experiment. V.T., R.H., and M.T. collected the data. M.T. and R.H. analyzed the data. M.T. wrote the manuscript. S.G. advised the study and secured funding.

## Competing interests

The authors declare no competing interests.
