## [Peer Review File · Nature Communications]

Reviewers' comments:

Reviewer #1 (Remarks to the Author):

This manuscript reports some interesting fMRI results for a behavioral paradigm and computational model that was already published by the same authors. In essence, the paper shows a neural dissociation between directed and random exploration, which rely on frontopolar versus dorsolateral prefrontal cortex, respectively. While these findings dovetail with similar results reported by Badre and colleagues, a novel and interesting result reported here is that adding the neural data from the regions to a choice model improves the prediction of behavior.

Here is a list of issues the authors should address in a revision:

(1) It is unclear from the description in the manuscript whether the behavioral models and the regressors in the neural GLM really account for the change in the observer's uncertainty from trial-to-trial. In other words, are RU and TU (e.g., in equation 4) indeed posteriors that are updated trial-by-trial, or are they set to the level of uncertainty inherent in the design of the different blocks? And are the parametric modulators in the neural GLM identical with these model-derived quantities? This was not clear since different models used different notation. Please clarify this for every model when you describe it in the text.

(2) The authors first identify regions of interest at uncorrected statistical thresholds, and then extract betas from these regions that they analyze with other statistical test. This circular "double-dipping" is problematic – of course the ROI analyses will confirm the significance of the analyses used to select the regions for extraction. The authors should either rely on SPM for their statistics (properly correcting for multiple comparisons) and only display the betas for visualization, or they should define their ROIs with fully independent criteria, extract betas, and then perform the statistical analyses only on these betas. If the authors lack sensitivity for whole-brain correction of the SPMs, then they could restrict their search (and correction) to a search mask comprising anatomical areas (not specific ROIs, but general areas) identified by prior studies. This mask should obviously have been selected a priori, but it would still be better for the field if the manuscript reported the results of such a non-circular strategy with proper correction for multiple comparisons.

(3) It is a very interesting result that adding the neural data improves choice prediction. However, the authors only use likelihood ratio tests to demonstrate this improvement. Critics may argue that it is not surprising that model fits improve if one adds any additional regressor to the model (overfitting). To counter such criticism pre-emptively, the authors should add other model comparison indices that penalize for the increase in model complexity, and discuss whether adding the neural data really improves the model when accounting for increased complexity, or whether it only increases the data fit.

(4) The authors only report regions that show a positive correlation with their uncertainty indices, arguing that negative correlations would not be consistent with the rate code implicit in their model. I

do not share this view, since their model does not really make any assumptions about neural coding of the computed quantities. Moreover, it is known from single-unit-recording studies of value-based choice that both positive and negative coding of choice-related representations are common (see e.g. Padoa-Schioppa's studies). The authors should therefore analyze and report the regions showing any correlation with their model-derived uncertainty representations.

Reviewer #2 (Remarks to the Author):

Review of manuscript NCOMMS-19-04648

Dissociable neural correlates of uncertainty underlie different exploration strategies by Tomov et al.

In this manuscript, the authors investigate the neural correlates of two different types of uncertainty, relative and total uncertainty of decision options, and their relation to choice in the context of an exploration/exploitation task. Understanding the neural processes associated with exploration-exploitation decisions is an important goal in the study of the neuroscience of decision-making, and the present manuscript makes an important contribution to the field. Below are some comments and concerns I believe the authors should address.

(1) The model of choices (eq. 4) is 'reduced-form' in that it relates choice to a linear combination of different types of uncertainty. Thus, the model is a statistical description of choices in terms of a simple combination of different types of uncertainty. However, the model doesn't explain how decision-makers decide which strategy, directed and random exploration, to use in a given decision situation. I would like the authors to address the latter question and describe what optimal behavior would look like of an agent who employed both of those strategies. Without such an explanation, the analysis is merely a 'curve-fitting' exercise and doesn't provide much insight into the decision-making process.

(2) The authors' computational model postulates that explore-exploit decisions are affected by two different types of computational strategies. In the manuscript (l. 51), the authors state that the two strategies confer different computational advantages, which, I believe, the authors mean to be advantages distinct from ecological advantages. The authors should explain what these advantages are. They should also relate the computational properties of the strategies (e.g., type of computation, memory requirements) to candidate neural processes/circuits.

(3) What is the covariance between the different IVs in the behavioural model (eq. 4)?

(4) As the authors point out, the exploration-exploitation problem is intractable in the general case. The authors should at least comment on the tractability of the computations necessary to implement their proposed model.

(5) In the fMRI analysis, the authors use a cluster extent threshold of 100 voxels (e.g., l. 144). I would like to see some justification of this choice. It appears somewhat arbitrary. Similarly, at various points in the fMRI analysis, the authors state that they Bonferroni corrected for 10 comparisons (e.g., l. 151) – the authors should mention how they choose 10.

(6) On l. 255, the authors state that participants received a variable payment based on a randomly drawn trial, and that losses were converted into a reward of 1. Were participants aware that this would happen (i.e., that they couldn't lose any money)? If this was the case, it appears that this would have fundamentally altered the incentive structure of the task. Participants were effectively holding a put option, and it appears that their optimal strategy would have been to maximise reward variance. I would appreciate if they authors could comment on this and how it may have affected their results.

(7) Overall, I find the fMRI analysis rather cursory. The authors state in the Summary that their results are “consistent with a hybrid computational architecture in which different uncertainty computations are performed separately and then combined by downstream decision circuits to compute choice”. Yet, all the authors do in the manuscript is present a set of neural correlates of variables capturing different types of uncertainty. There is very little discussion relating their findings to the statement above, in particular, how their findings support the claim of a “hybrid computational architecture”, let alone what this architecture would look like (e.g., neural circuits involved, anatomical properties of implicated regions), or how the different computations might be “combined by downstream decision circuits to compute choice”. In fact, I didn't see much evidence in the manuscript supporting the latter claim at all. For example, the authors could have provided some evidence of connectivity between the regions coding for different types of uncertainty and the regions computing choice (or DCM?), to provide tighter evidence that the computations of uncertainty did in fact affect choice and to what extent.

Reviewer #3 (Remarks to the Author):

Using fMRI, The paper analyses human brain activations in two armed-bandit tasks comprising two risky arms, two safe arms or one safe and one risky arm. The analysis is based on a Kalman filter used to model subjects' behavior. The paper reports that relative uncertainty between arms is associated with rostralateral prefrontal activations, while left dorsolateral prefrontal activations correlate with total uncertainty of the bandit. Overall, I don't see what we learn new from this data set. Moreover, there are major drawbacks in the Methods, which do not meet current methodological standards in the field.

Critical concerns:

1/ The results are essentially a replication of those from Badre et al. (2012) recast in the framework of Kalman Filters. The only difference between the present and Badre et al.'s results is that the present study reports left DLPFC activations while Badre et al. reported right DLPFC activations correlating with total uncertainty of choice options. However, the paper provide no explanations about this discrepancy

which questions the replicability and generalization of this finding.

2/ the behavioral and neural data reported in the paper actually provide evidence against the proposed Kalman filter model used in analyzing behavioral and fMRI data. FigS2 shows that the model behavior significantly and strongly differs from subjects' performances. Additionally, the variables V and V/TU, which captures significant parts of the behavioral variance beyond RU were associated with no brain activations. Moreover, the study comprises no model comparisons. Model comparisons is the current standard in model-based fMRI studies and is considered as necessary for drawing conclusions about the underlying cognitive and brain processes (see e.g. Palminteri et al., 2017, TICS).

3/ The neural decoding analysis is based on comparing likelihood ratios using T-tests. However, this analysis is statistically biased as the neurally augmented model likelihood is by construction larger than the behavioral model likelihood (as including more free parameters). This analysis should be based on the BICs, which may change the results as the reported p-value is close to the significance threshold ($p=0.04$).

Reviewer #1 (Remarks to the Author):

(1) It is unclear from the description in the manuscript whether the behavioral models and the regressors in the neural GLM really account for the change in the observer's uncertainty from trial-to-trial. In other words, are RU and TU (e.g., in equation 4) indeed posteriors that are updated trial-by-trial, or are they set to the level of uncertainty inherent in the design of the different blocks? And are the parametric modulators in the neural GLM identical with these model-derived quantities? This was not clear since different models used different notation. Please clarify this for every model when you describe it in the text.

We now state explicitly that those are trial-by-trial posteriors derived from the model, and we include trial (t) subscripts everywhere to make that clear. We also clarify that the parametric modulators were the same posterior quantities. We also ensured that the notation is consistent.

(2) The authors first identify regions of interest at uncorrected statistical thresholds, and then extract betas from these regions that they analyze with other statistical test. This circular "double-dipping" is problematic – of course the ROI analyses will confirm the significance of the analyses used to select the regions for extraction. The authors should either rely on SPM for their statistics (properly correcting for multiple comparisons) and only display the betas for visualization, or they should define their ROIs with fully independent criteria, extract betas, and then perform the statistical analyses only on these betas. If the authors lack sensitivity for whole-brain correction of the SPMs, then they could restrict their search (and correction) to a search mask comprising anatomical areas (not specific ROIS, but general areas) identified by prior studies. This mask should obviously have been selected a priori, but it would still be better for the field if the manuscript reported the results of such a non-circular strategy with proper correction for multiple comparisons.

We now report corrected results and use the *a priori* ROIs from Badre et al. (2012) in our confirmatory analyses.

(3) It is a very interesting result that adding the neural data improves choice prediction. However, the authors only use likelihood ratio tests to demonstrate this improvement. Critics may argue that it is not surprising that model fits improve if one adds any additional regressor to the model (overfitting). To counter such criticism pre-emptively, the authors should add other model comparison indices that penalize for the increase in model complexity, and discuss whether adding the neural data really improves the model when accounting for increased complexity, or whether it only increases the data fit.

We now base our comparisons on BIC's, which more stringently penalize model complexity. Also, we should note that the likelihood ratio test does penalize model complexity (albeit less stringently than BIC), so it is not the case that adding additional regressors will necessarily improve the likelihood ratio.

(4) The authors only report regions that show a positive correlation with their uncertainty indices, arguing that negative correlations would not be consistent with the rate code implicit in their model. I do not share this view, since their model does not really make any assumptions about neural coding of the computed quantities. Moreover, it is known from single-unit-recording studies of value-based choice that both positive and negative coding of choice-related representations are common (see e.g. Padoa-Schioppa's studies). The authors should therefore analyze and report the regions showing any correlation with their model-derived uncertainty representations.

We now report and analyze negative as well as positive correlations.

Reviewer #2 (Remarks to the Author):

(1) The model of choices (eq. 4) is 'reduced-form' in that it relates choice to a linear combination of different types of uncertainty. Thus, the model is a statistical description of choices in terms of a simple combination of different types of uncertainty. However, the model doesn't explain how decision-makers decide which strategy, directed and random exploration, to use in a given decision situation. I would like the authors to address the latter question and describe what optimal behavior would look like of an agent who employed both of those strategies. Without such an explanation, the analysis is merely a 'curve-fitting' exercise and doesn't provide much insight into the decision-making process.

The model is not reduced form, because there isn't a more complex model that it is reducing. If one takes only the $V + RU$ terms (excluding the V/TU term), then the model precisely specifies the choice probability under the UCB exploration policy. If one takes on the V/TU term, then the model precisely specifies the choice probability under the Thompson sampling policy. The only somewhat arbitrary aspect here is when we combine UCB and Thompson sampling models via addition. However, we have shown previously that this hybrid model is able to capture the choice probability functions quite accurately (see for example Gershman & Tzovaras, 2018). So it doesn't seem necessary to further explore the space of hybrid models in this paper. Our focus here is less on how they combine and more on how the underlying components are represented.

With regard to the meta-choice of which strategy to use, we have assumed that the same linear combination applies to every trial (hence there is no dynamic meta-strategy). This assumption may be wrong, and exploring alternative assumptions would be a very interesting question for future work, but outside the scope of this paper. Nonetheless, to address the question of optimality, we performed simulations to demonstrate that the hybrid UCB/Thompson model is superior to UCB or Thompson sampling alone, which are in turn superior to softmax alone both in terms of performance and in terms of describing human behavior.

We added the following to the paper (p. 10):

Gershman (2018) showed that, despite its apparent simplicity, this is not a reduced form model but rather the exact analytical form of the most parsimonious hybrid of UCB and Thompson sampling that reduces to pure UCB when $w_3 = 0$, to pure Thompson sampling when $w_2 = 0$, and to pure softmax exploration when $w_2 = w_3 = 0$. Thus the hybrid model balances exploitation (governed by w_1) with directed (w_2) and random (w_3) exploration simultaneously

for each choice, without the need to dynamically select one strategy over the other (whether and how the brain might perform this meta-decision is beyond the scope of our present work). If subjects use both UCB and Thompson sampling, the model predicts that all three regressors will have a significant effect on choices ($w_1 > 0, w_2 > 0, w_3 > 0$).

Correspondingly, the maximum likelihood estimates of all three fixed effects coefficients were significantly greater than zero: $w_1 = 0.166 \pm 0.016$ ($t(9716) = 10.34, p = p < 10^{-20}$; mean \pm s.e.m., two-tailed t-test), $w_2 = 0.175 \pm 0.021$ ($t(9716) = 8.17, p = p < 10^{-15}$), and $w_3 = 0.005 \pm 0.001$ ($t(9716) = 4.47, p = p < 10^{-5}$). Model comparisons revealed that the UCB/Thompson hybrid model fits subject choices better than UCB or Thompson sampling alone, which in turn fit choices better than softmax alone (Table S1). Furthermore, running these models generatively with the corresponding fitted parameters on the same bandits as the participants revealed significant differences in model performance (Figure S3, $F(3, 1236) = 291.58, p < 10^{-20}$, one-way ANOVA). The UCB/Thompson hybrid outperformed UCB and Thompson sampling alone (UCB vs. hybrid, $p < 10^{-8}$; Thompson vs. hybrid, $p < 10^{-8}$, pairwise multiple comparison tests), which in turn outperformed softmax exploration (softmax vs. UCB, $p < 10^{-5}$; softmax vs. Thompson, $p < 10^{-8}$). Similar results replicated across a range of coefficients (Figure S4), signifying the distinct and complementary ecological advantages of UCB and Thompson sampling. Thus relying on both UCB ($w_2 > 0$) and Thompson sampling ($w_3 > 0$) should yield better overall performance. In line with this prediction, we found better performance among subjects whose choices are more sensitive to RU_t (greater w_2), consistent with greater reliance on UCB (Figure S5B, $r(29) = 0.47, p = 0.008$, Pearson correlation). Similarly, we found better performance among subjects whose choices are more sensitive to V_t/TU_t (greater w_3), consistent with greater reliance on Thompson sampling (Figure S5C, $r(29) = 0.53, p = 0.002$). Finally, note that even though optimal exploration is intractable in general, the hybrid model computes choices in constant time by simply computing Eq. 4. Taken together, these results replicate and expand upon previous findings (Gershman, 2019), highlighting the superiority of the UCB/Thompson hybrid as a descriptive as well as normative model of uncertainty-guided exploration. Thus humans do and ought to employ both directed and random exploration, driven by relative and total uncertainty, respectively.

(2) The authors' computational model postulates that explore-exploit decisions are affected by two different types of computational strategies. In the manuscript (l. 51), the authors state that the two strategies confer different computational advantages, which, I believe, the authors mean to be advantages distinct from ecological advantages. The authors should explain what these advantages are. They should also relate the computational properties of the strategies (e.g., type of computation, memory requirements) to candidate neural processes/circuits.

We do in fact mean ecological advantages, which we hope the above paragraph illustrates. We also corrected "computational advantages" to "ecological advantages" in the text.

(3) What is the covariance between the different IVs in the behavioural model (eq. 4)?

The correlations are as follows (Pearson correlation computed for each subject; then t-test performed with Fisher z-transformed r values):

V vs. RU: $r = -0.19 \pm 0.03$, $t(30) = -6.78$, $p < 10^{-6}$
V vs. TU: $r = 0.06 \pm 0.02$, $t(30) = 3.57$, $p = 0.0012$
RU vs. TU: $r = -0.10 \pm 0.03$, $t(30) = -3.24$, $p = 0.002$

The correlations are small but significant, which in principle could pose a problem for interpreting the coefficients w , since it is possible that they trade off with each other. We believe this is not the case, as illustrated by several of our results:

1. Figure S6 clearly illustrates the parameter recoverability of our fitting procedure even in the presence of such correlations.
2. Figure S5 shows that the subject-specific coefficients are related to performance, suggesting that meaningful coefficients were obtained despite the correlations.
3. The behavioral model comparison (Table S1) and performance comparison (Figure S3) show that the hybrid model with the fitted coefficients is superior, which would not have been the case if the coefficients were not fit in a way that takes advantage of all three regressors and hence both random and directed exploration.
4. The VIF analysis in the supplemental information (Figure S8) address this concern with regards to the fMRI analysis and shows that, despite the correlations, including all regressors in the same GLM (GLM 1) yields essentially

the same BOLD activations as the single-parametric modulator GLMs (Figure S9).

(4) As the authors point out, the exploration-exploitation problem is intractable in the general case. The authors should at least comment on the tractability of the computations necessary to implement their proposed model.

The model computes choices in constant time, which we clarify in the above paragraph.

(5) In the fMRI analysis, the authors use a cluster extent threshold of 100 voxels (e.g., l. 144). I would like to see some justification of this choice. It appears somewhat arbitrary. Similarly, at various points in the fMRI analysis, the authors state that they Bonferroni corrected for 10 comparisons (e.g., l. 151) – the authors should mention how they choose 10.

We agree with the reviewer that the cluster extent threshold was somewhat arbitrary. It was originally based on strong prior hypothesis that right RLPFC would encode relative uncertainty, based on Badre et al. (2012)'s results. We have now removed the arbitrary extent threshold and report whole-brain results with cluster FWE correction as well as ROI results with *a priori* ROIs.

We previously used Bonferroni correction for the 10 ROIs we identified in our whole-brain contrast. However, since now we focus our confirmatory analyses on the *a priori* ROIs only, we no longer correct for multiple ROIs.

(6) On l. 255, the authors state that participants received a variable payment based on a randomly drawn trial, and that losses were converted into a reward of 1. Were participants aware that this would happen (i.e., that they couldn't lose any money)? If this was the case, it appears that this would have fundamentally altered the incentive structure of the task. Participants were effectively holding a put option, and it appears that their optimal strategy would have been to maximise reward variance. I would appreciate if they authors could comment on this and how it may have affected their results.

Participants were aware of this. While it is true that participants were effectively holding a call option, they nevertheless behaved as if they sought to maximize reward, as can be seen in Figure S1A. We added the following (p. 23):

While eliminating the possibility of losses may appear to have altered the incentive structure of the task, participants nevertheless preferred the better option across all task conditions (Figure S1A), in accordance with previous replications of the experiment (Gershman, 2018; 2019).

(7) Overall, I find the fMRI analysis rather cursory. The authors state in the Summary that their results are “consistent with a hybrid computational architecture in which different uncertainty computations are performed separately and then combined by downstream decision circuits to compute choice”. Yet, all the authors do in the manuscript is present a set of neural correlates of variables capturing different types of uncertainty. There is very little discussion relating their findings to the statement above, in particular, how their findings support the claim of a “hybrid computational architecture”, let alone what this architecture would look like (e.g., neural circuits involved, anatomical properties of implicated regions), or how the different computations might be “combined by downstream decision circuits to compute choice”. In fact, I didn’t see much evidence in the manuscript supporting the latter claim at all. For example, the authors could have provided some evidence of connectivity between the regions coding for different types of uncertainty and the regions computing choice (or DCM?), to provide tighter evidence that the computations of uncertainty did in fact affect choice and to what extent.

We now include an entirely new section in the results with a new GLM (GLM 2) showing neural correlates of the decision value (DV) in motor cortex, suggesting that it performs the downstream decision computation that combines V, RU, and TU. Note that since DV is by design correlated with V, RU, and V/TU, this also implies the region is functionally coupled with the RU and TU ROIs.

Reviewer #3 (Remarks to the Author):

Critical concerns:

1/ The results are essentially a replication of those from Badre et al. (2012) recast in the framework of Kalman Filters. The only difference between the present and Badre et al.’s results is that the present study reports left DLPFC activations while Badre et al. reported right DLPFC activations correlating with total uncertainty of choice options. However, the paper provide no explanations about this discrepancy which questions the replicability and generalization of this finding.

We appreciate the reviewer’s concerns, and while our work indeed replicates the results of Badre et al. (note that after correcting the bug in our pipeline, we now replicate the TU result in right DLPFC as well), it goes beyond it in several important ways:

1. Our experimental design explicitly orthogonalizes RU and TU, thus providing a cleaner test of the hypothesis,

2. Our UCB/Thompson hybrid model explains how TU is used by the brain,
3. We show that neural activity predicts behavioral variance, and
4. We investigate the downstream circuitry which combines RU and TU to compute choice.

None of these questions were addressed in the original Badre et al. paper.

We clarified this in the discussion (p. 18):

computations predicted by the model was reified as an anatomical dissociation between their neural correlates. Our GLM results confirm the previously identified role of right RLPFC and right DLPFC in encoding relative and total uncertainty, respectively (Badre et al., 2012). Crucially, our work further elaborates the functional role of those regions by providing a normative account of how both uncertainty estimates are used by the brain to make choices, with relative uncertainty driving direct exploration and total uncertainty driving random exploration. This account was validated by our decoding analysis and decision value GLM, which suggest that the two uncertainty estimates are combined with the value estimate in downstream motor cortex, which ultimately performs the categorical decision computation.

2/ the behavioral and neural data reported in the paper actually provide evidence against the proposed Kalman filter model used in analyzing behavioral and fMRI data. FigS2 shows that the model behavior significantly and strongly differs from subjects' performances. Additionally, the variables V and V/TU, which captures significant parts of the behavioral variance beyond RU were associated with no brain activations. Moreover, the study comprises no model comparisons. Model comparisons is the current standard in model-based fMRI studies and is considered as necessary for drawing conclusions about the underlying cognitive and brain processes (see e.g. Palminteri et al., 2017, TICS).

We are not sure which aspects of the figure the reviewer is referring to as “significantly and strongly” differing from subject behavior. We believe our results show that the model captures the main qualitative patterns of human behavior relevant to our study, as explained in the results section (p. 9, 10, 11) and as evident in Figures S1, S2, and S5. It also quantitatively accounts for behavior better than alternative models (Table S1). Note that these behavioral results have been replicated with greater sample sizes on the same task (Gershman 2019; Gershman & Tzovaras, 2018).

We discuss the variable V in the supplemental information (p. 39). We added the following regarding the absence of V/TU in the discussion:

Another puzzling aspect of our results that merits further investigation is the lack of any signal corresponding to V_t/TU_t . This suggests that the division is performed by circuits downstream from right DLPFC, such as motor cortex. Alternatively, it could be that, true to Thompson sampling, the brain is generating samples from the UCB-adjusted posterior value distributions and comparing them to make decisions. In that case, what we are seeing in motor cortex could be the average of those samples, consistent with the analytical form of the UCB/Thompson hybrid (Eq. 4) which is derived precisely by averaging over all possible samples (Gershman, 2018). Further studies could disambiguate between these alternatives and investigate the precise mechanism of random exploration and how it is implemented by neural circuits.

We also included model comparisons (Table 1, Table S1). If the reviewer is referring to neural model comparisons, we did not perform any as none of the alternative models provided an equally compelling account of behavior (Table S1).

3/ The neural decoding analysis is based on comparing likelihood ratios using T-tests. However, this analysis is statistically biased as the neurally augmented model likelihood is by construction larger than the behavioral model likelihood (as including more free parameters). This analysis should be based on the BICs, which may change the results as the reported p-value is close to the significance threshold (p=0.04).

We now use BICs for model comparison. Note that since we fixed a bug in our pipeline and re-analyzed the data, the differences in model fits are greater. Please also note that the likelihood ratio test does penalize model complexity (albeit less stringently than BIC), so it is not the case that adding additional regressors will necessarily improve the likelihood ratio.

Reviewers' comments:

Reviewer #1 (Remarks to the Author):

The authors have successfully addressed my previous concerns, and they have strengthened their manuscript by additional analyses revealing some downstream value integration in left motor cortex. While it is not entirely clear why they observe this integration there rather than in, say, ventromedial cortex, they discuss this issue in a manner that will satisfy most readers. It may still be good to point out in the discussion that for these right-handed subjects, left motor cortex is the final cortical area implementing the motor choice. This suggests that it may be interesting for future studies to investigate whether the accumulation area will shift if subjects give their response with another motor modality (eye movements, left hand, etc). Such a qualifying statement would render the result more consistent with the existing literature and would circumvent the impression that left motor cortex is a general value accumulator in many different contexts.

I also have two other minor comments concerning results reporting: In the legend of Figure 3, the authors should explicitly state whether all displayed voxels are also corrected at cluster-level (as written in the text) or only at the voxel-level? In the legends of Figures 4a and 5 a, they should also report the statistical threshold procedures used for display (the statement about identical procedures in the legend of Panel 4b is ambiguous in that respect).

Reviewer #2 (Remarks to the Author):

The authors have addressed all of my comments to my satisfaction.

Reviewer #3 (Remarks to the Author):

I carefully read the revised paper along with the authors' responses to my initial major concerns. Except their revision including proper model comparisons in response to my point regarding the lack of such analyses, the authors have provided no convincing (and even meaningful I must say) responses/revisions to my other major concerns, as detailed below. Moreover, the authors report a bug in their initial analyses, indicating that the bug correction altered the results without detailing the alterations, except precisely those suppressing the discrepancy between their results and those in Badre et al. (2012). This leads to a very uncomfortable situation, in which the reviewer cannot be really trustful in the presented results.

Major concerns:

Initial major point (1):

The authors' response confirms my initial judgment: the results are mainly a replication of those in

Badre et al. (2012) and the analyses carried out in the present paper provide no new insights. In their response, the authors only state that their results go beyond Badre et al.'s results because:

a/ "Our experimental design explicitly orthogonalizes RU and TU, thus providing a cleaner test of the hypothesis".

This is inexact: Badre et al. (2012) also orthogonalized RU et TU.

b/ "Our UCB/Thompson hybrid model explains how TU is used by the brain".

This is also incorrect : the UCB/Thompson hybrid model predicts brain activations correlating with V/TU, but no such activations were found.

c/ "We show that neural activity predicts behavioral variance"

These analyses are welcomed but are purely confirmatory without providing new insights. (see also my comment below in point 3)

d/ "We investigate the downstream circuitry which combines RU and TU to compute choice".

As reported in the paper, the additional finding in the motor cortex added in the revised paper remain anecdotal: the related analysis does not include reaction times as a confound factor. Including RTs as a factor of no-interest is the current standard in fMRI to assess activations associated with the decision variable. Additionally and possibly relatedly, no previous studies report motor activations associated with the decision variable. Previous studies report such activations in the medial PFC, lateral PFC or in the parietal cortex . Moreover, the reported analysis arbitrary focused on positive correlations, whereas the meaningful correlation reported in previous studies is the negative correlation between brain activations and the decision variable.

Initial major point (2):

My original concern was that the reported results seem to rule out rather than to support the authors' model. Regarding behavioral data, it is clear that in Fig. S2, there are significant differences between participants' and model's learning curves: from trial 1 to trial 2 in condition SS, subjects' performances decreased, whereas the model performance increased, with a significant difference between the two curves. Moreover, in conditions RR, RS and SR, subjects reached a plateau at ~90% performance, whereas the model reached a clearly significant lower plateau at ~80% performance. The authors inappropriately and unconvincingly responded that they only "believe" the model "qualitatively" accounts for the data!!! Regarding fMRI data, the authors acknowledge that they found no activations associated with V and V/TU, which are actually two critical predictions from the proposed model. For V, they reported additional analyses, which are however fully inconclusive as not controlling for co-linearity effects across regressors. As a reader, I still conclude from the revised paper that problematically, the reported data set appears to rule out rather than support the proposed model.

Initial major point (3)

As requested, the author conducted the proper model comparisons in the revised paper based on BICs, but missed to report the related statistical tests. They need to report paired T-tests or to use an exceedance probability approach to compare BICs between models.

Reviewer #1 (Remarks to the Author):

The authors have successfully addressed my previous concerns, and they have strengthened their manuscript by additional analyses revealing some downstream value integration in left motor cortex. While it is not entirely clear why they observe this integration there rather than in, say, ventromedial cortex, they discuss this issue in a manner that will satisfy most readers. It may still be good to point out in the discussion that for these right-handed subjects, left motor cortex is the final cortical area implementing the motor choice. This suggests that it may be interesting for future studies to investigate whether the accumulation area will shift if subjects give their response with another motor modality (eye movements, left hand, etc). Such a qualifying statement would render the result more consistent with the existing literature and would circumvent the impression that left motor cortex is a general value accumulator in many different contexts.

I also have two other minor comments concerning results reporting: In the legend of Figure 3, the authors should explicitly state whether all displayed voxels are also corrected at cluster-level (as written in the text) or only at the voxel-level? In the legends of Figures 4a and 5 a, they should also report the statistical threshold procedures used for display (the statement about identical procedures in the legend of Panel 4b is ambiguous in that respect).

We have incorporated the requested changes in the manuscript.

Reviewer #2 (Remarks to the Author):

The authors have addressed all of my comments to my satisfaction.

Reviewer #3 (Remarks to the Author):

I carefully read the revised paper along with the authors' responses to my initial major concerns. Except their revision including proper model comparisons in response to my point regarding the lack of such analyses, the authors have provided no convincing (and even meaningful I must say) responses/revisions to my other major concerns, as detailed below. Moreover, the authors report a bug in their initial analyses, indicating that the bug correction altered the results without detailing the alterations, except precisely those suppressing the discrepancy between their results and those in Badre et al. (2012). This leads to a very uncomfortable situation, in which the reviewer cannot be really trustful in the presented results.

The bug was in our univariate decoder code. Specifically, we were decoding a mix of V/TU and TU (due to the common suffix in the regressor names) instead of TU. Using this incorrectly decoded TU, we were unable to improve choice predictions using the BOLD signal in right DLPFC, the ROI reported by Badre et al. Instead, we obtained a significant result using the BOLD signal in left DLPFC within the ROIs discovered by our whole-brain TU contrast (this was using the likelihood ratio test, a less stringent criterion than the one used in the revision). After fixing the bug, we found the opposite pattern: decoding TU from left DLPFC does not improve choice predictions, but decoding it from right DLPFC does (also note that this time we were using BIC's). Note that this bug only affects our confirmatory analysis, in particular the one for TU only. It does not affect the GLM results, nor the confirmatory analysis for RU.

Major concerns:

Initial major point (1):

The authors' response confirms my initial judgment: the results are mainly a replication of those in Badre et al. (2012) and the analyses carried out in the present paper provide no new insights. In their response, the authors only state that their results go beyond Badre et al.'s results because:

a/ "Our experimental design explicitly orthogonalizes RU and TU, thus providing a cleaner test of the hypothesis".

This is inexact: Badre et al. (2012) also orthogonalized RU et TU.

We should have clarified that we mean that TU and RU are orthogonalized in the task design (and not merely in the fMRI GLM, which we assume the reviewer is referring to). We added the following to the discussion section:

"While our study replicates the results reported by Badre et al., it goes beyond their work in several important ways. First, our task design explicitly manipulates uncertainty – the main quantity of interest – across the different task conditions, whereas the task design in Badre et al. is focused on manipulating expected value. Second, relative and total uncertainty are manipulated independently in our task design: relative uncertainty differs across RS and SR trials, while total uncertainty remains fixed, on average; the converse holds for SS and RR trials. Orthogonalizing relative and total uncertainty in this way allows us to directly assess their differential contribution to choices (figure S2). Third, the exploration strategies employed by our model are rooted in normative principles developed in the machine learning literature (Thompson, Aurora), with theoretical performance guarantees which were confirmed by our simulations (Figure S3 and S4). In particular, the separate contributions of relative and total uncertainty to choices are derived directly from UCB and Thompson sampling, implementing directed and random exploration, respectively. Fourth, this allows us to link relative and total uncertainty and their neural correlates directly to subject behavior and interpret the results in light of the corresponding exploration strategies."

b/ "Our UCB/Thompson hybrid model explains how TU is used by the brain". This is also incorrect : the UCB/Thompson hybrid model predicts brain activations correlating with V/TU, but no such activations were found.

We should have clarified that what we mean is that the model provides a principled explanation of the role of TU in guiding choices, without committing to a particular mechanism. While it is puzzling that we found no results for V/TU, this is likely due to the particular mathematical form of the choice probability that we used in our analysis, which need not necessarily correspond to the underlying neural mechanism. For example, a sequential sampling scheme (e.g. a DDM) that samples from the UCB-adjusted posteriors would not predict explicit encoding of V/TU since it is not involved in the computations, yet it would still predict encoding of TU as part of storing and updating the posteriors, and importantly, it will make choice predictions that are consistent with our hybrid model (which, in contrast, does involve V/TU). For a detailed derivation, please refer to Gershman (2018).

We have the following paragraph in the discussion section:

"One mechanism by which this could occur is suggested by sequential sampling models, which posit that the decision value DV drives a noisy accumulator to a decision bound, at which point a decision is made (Busemeyer 1993). This is consistent with Gershman (2019)'s analysis of reaction time patterns on the same task as ours. It is also consistent with studies reporting neural signatures of evidence accumulation during perceptual as well as value-based judgments in human motor cortex (Gratton 1988, Graziano 2011, Hare 2011, Gluth 2012, Polania 2014). It is worth noting that for our right-handed subjects, left motor cortex is the final cortical area implementing the motor choice. One potential avenue for future studies would be to investigate whether the decision value area will shift if subjects respond using a different modality, such as their left hand, or using eye movements. This would be consistent with previous studies that have identified effector-specific value coding in human cortex (Gershman 2009)."

And we added the following text to the next paragraph:

"A sampling mechanism could thus explain both the negative sign of the $|DV|$ effect in motor cortex, as well as the absence of V/TU in the BOLD signal."

c/ "We show that neural activity predicts behavioral variance"
These analyses are welcomed but are purely confirmatory without providing new insights. (see also my comment below in point 3)

We believe that the confirmatory analyses provide an important link between the neural correlates of the uncertainty quantities and the corresponding exploration strategies, consistent with their theoretically predicted role in guiding behavior.

d/ "We investigate the downstream circuitry which combines RU and TU to compute choice".

As reported in the paper, the additional finding in the motor cortex added in the revised paper remain anecdotal: the related analysis does not include reaction times as a confound factor. Including RTs as a factor of no-interest is the current standard in fMRI to assess activations associated with the decision variable..

We included the following analysis in the Supplemental Information section:

“One potential confound of our decision value result (GLM 2) in motor cortex is reaction time (RT). When including RT's as a parametric modulator in addition to DV (GLM 2A), we found no effect of DV in motor cortex (no voxels survived cluster FWE correction). Note, however, that the sequential sampling framework predicts a strong relationship between DV and RT's: when DV is close to zero, the two options are similar to each other and hence it takes longer for the evidence accumulator to reach a decision bound. This prediction was manifested in our data (coefficient = -0.006, $F(1,9717) = 23.8$, $p = 0.000001$, mixed effects linear regression: $RT \sim 1 + DV + (1 + DV | \text{SubjectID})$), indicating that the negative result could be due to RT's capturing some of the shared variance in the BOLD signal.

To account for this possibility, we performed random effects Bayesian model comparison (Rigoux 2014) between the GLM with DV alone (GLM 2), the GLM with both DV and RT (GLM 2A), and a GLM with RT alone (GLM 2B) in the left motor cortex ROI identified by GLM 2 (Figure 5A). Specifically, following our previous work (Tomov 2018), we approximated the log model evidence as $-0.5 * \text{BIC}$, where the BIC was computed based on the residual variance of the GLM fits within a 10 mm sphere around the peak voxel in left motor cortex from GLM 2 (MNI [-38 -8 62]). This analysis strongly favored GLM 2A (PXP = 0.96) over GLM 2 (PXP = 0) and GLM 2B (PXP = 0.04). This indicates that the BOLD signal in left primary motor cortex is best explained by combination of DV and RT, rather than RT or DV alone, pointing to decision value coding in left primary motor cortex above and beyond RT's.”

Additionally and possibly relatedly, no previous studies report motor activations associated with the decision variable. Previous studies report such activations in the medial PFC, lateral PFC or in the parietal cortex

We added the following to our discussion:

“However, most of these studies consider the values of the available options (Q) or the difference between them (V), without taking into account the uncertainty of those quantities. This suggests that the values encoded in those regions are divorced from any uncertainty-related information, which would render them insufficient to drive uncertainty-guided exploratory behavior on their own. Uncertainty would have to be computed elsewhere and then integrated with these value signals by downstream decision circuits closer to motor output”

Moreover, the reported analysis arbitrary focused on positive correlations, whereas the meaningful correlation reported in previous studies is the negative correlation between brain activations and the decision variable.

In the revision, we already report both positive and negative contrasts (Fig 5, S7, S9; Tab S3, S4, S5). Note that the DV effect in motor cortex is in fact negative, consistent with the previous studies, which we highlight in the discussion.

Initial major point (2):

My original concern was that the reported results seem to rule out rather than to support the authors' model. Regarding behavioral data, it is clear that in Fig. S2, there are significant differences between participants' and model's learning curves: from trial 1 to trial 2 in condition SS, subjects' performances decreased, whereas the model performance increased, with a significant difference between the two curves.

The difference from trial 1 to trial 2 is not significant for the subjects or the model:

SS trial 1 vs. trial 2, human: $t(30) = 0.802$, $p = 0.4290$

SS trial 1 vs. trial 2, model: $t(30) = -1.517$, $p = 0.1398$

Moreover, in conditions RR, RS and SR, subjects reached a plateau at ~90% performance, whereas the model reached a clearly significant lower plateau at ~80% performance. The authors inappropriately and unconvincingly responded that they only “believe” the model “qualitatively” accounts for the data!!!

Our model does reach a lower asymptotic performance compared to the participants. This could be due to model parameters that were not fit to choices, e.g. a discrepancy between our prior variances and the subjects', causing them to decrease their uncertainty faster and start exploiting sooner. Since our primary interest is in exploratory behavior rather than asymptotic performance, we focus on the key parameters of the model that arbitrate between the different exploration strategies (the w 's). We would like to highlight that the goal

of the model is not to perfectly capture every aspect of behavior (something which is beyond the scope of any model) but to explain the structure of exploration algorithms that the brain is using. We capture this structure quantitatively (Tab S1), not just qualitatively, consistent with previous work in our lab (Gershman 2018, Gershman & Tzovaras 2018, Gershman 2019), which shows that these modeling results are robust.

Regarding fMRI data, the authors acknowledge that they found no activations associated with V and V/TU, which are actually two critical predictions from the proposed model.

We discuss the absence of V/TU in the discussion section:

“Another puzzling aspect of our results that merits further investigation is the lack of any signal corresponding to V/TU. This suggests that the division is performed by circuits downstream from right DLPFC, such as motor cortex. Alternatively, it could be that, true to Thompson sampling, the brain is generating samples from the UCB-adjusted posterior value distributions and comparing them to make decisions. In that case, what we are seeing in motor cortex could be the average of those samples, consistent with the analytical form of the UCB/Thompson hybrid, which is derived precisely by averaging over all possible samples (Gershman 2018). A sampling mechanism could thus explain both the negative sign of the DV effect in motor cortex, as well as the absence of V/TU in the BOLD signal. Further studies could disambiguate between these alternatives and investigate the precise mechanism of random exploration and how it is implemented by neural circuits.”

For V, they reported additional analyses, which are however fully inconclusive as not controlling for co-linearity effects across regressors. As a reader, I still conclude from the revised paper that problematically, the reported data set appears to rule out rather than support the proposed model.

Since our primary interest was uncertainty-driven exploration, we focused our task design on maximizing power for the uncertainty-related quantities in our model (V, RU, V/TU). Our VIF analysis (Fig S8) shows that indeed, V is correlated with the other regressors, which reduces power for V when controlling for the other regressors since they capture some of the shared variance. Hence the fact that we did not find any regions for V when controlling for the other regressors highlights a weakness of our experimental design, but does not necessarily rule out our model. Including V alone (as is done in many other studies which do not consider the role of uncertainty in decision-making and are not plagued by this problem, simply because they lack these confounding regressors) is the only way to circumvent this issue and the fact that this produces a result consistent with the prior literature is reassuring.

We added the following to the Supplemental Information section:

“ROI analysis using an anatomically defined vmPFC region (as a conjunction of Superior frontal gyrus, medial orbital; Superior frontal gyrus, medial; and Gyrus rectus from the AAL2 atlas) showed a significant positive effect of V in left vmPFC ($t(30) = 2.14$, $p = 0.04$, t-test of ROI-averaged betas across subjects).”

Initial major point (3)

As requested, the author conducted the proper model comparisons in the revised paper based on BICs, but missed to report the related statistical tests. They need to report paired T-tests or to use an exceedance probability approach to compare BICs between models.

We fitted a fixed effects version of the model separately for each subject and computed the protected exceedance probabilities (PXPs), which show that our model is superior to the alternatives ($PXP = 1$). Note that for a mixed effects analysis (which we use in the rest of our paper) we cannot compute single-subject BICs because they are coupled, and for the same reason we cannot compute PXPs. The Bayesian interpretation of the group-level BIC is that it is proportional to an approximation of the negative log marginal likelihood, which directly quantifies the model evidence (refer to Kass & Raftery, 1995). We added the following to our results section:

"Bayesian model comparison strongly favored the hybrid model over alternative models (protected exceedance probability = 1, Rigoux et al. 2014)"

And the following to our method section:

“For Bayesian model comparison, we fit w separately for each subject using fixed effects maximum likelihood estimation in order to obtain a separate BIC for each subject. We approximated the log model evidence for each subject as $-0.5 * BIC$ and used it to compute the protected exceedances probability for each model, which is the probability that the model is most prevalent in the population (Rigoux et al., 2014).”

Reviewers' comments:

Reviewer #3 (Remarks to the Author):

I appreciate the authors' efforts to respond to my previous reviews and the details they provide regarding the initial bug in their analyses. The present study uses a behavioral protocol dissociating total and relative uncertainty, while in Badre et al. (2012) these variables are orthogonalized in the regression analyses. The present study reports activations replicating those previously reported in Badre et al. (2012). The authors then claim in the paper that their study provides new insights as the results are derived from a normative adaptive model predicting activations associated with V/TU and reveal motor activations associated with the decision variable. However, the revised paper along with the additional analyses regarding motor activations still provides clear evidence against the authors' claims:

A- A first major problem is that in contradiction with the prediction, they found no activations associated with V/TU. The authors then explain in the discussion that this variable might not be explicitly encoded in neural networks implementing sequential sampling models. If the authors believe so and if they want their paper to go beyond Badre et al. (2012) and beyond a "wishful thinking", as they claim, they have to carry out the analyses comprising sequential sampling regressors rather than V/TU

B- A second major, even more critical problem is that the additional analyses reported in the revised supplementary information show the evidence that in contradiction with authors' claims, motor activations are actually unrelated to the Decision Variable (DV) but associated with Reaction Times (RT). When controlling for the confounding factor RT, as I requested in my previous review and as systematically performed in previous studies investigating neural correlates of DVs, the authors found that in the regression analysis, motor activations were unassociated with variable DV. The conclusion should be that consistent with the literature, motor activations reflect motor responses instead of decision variable DV. However, to overcome this problem, the authors carried out a notoriously, statistically biased analysis. The analysis falls into the double-dipping fallacy (Kriegeskorte et al., Nature Neuroscience, 2009) as performed on data selected within the cluster associated with DV according to the regression analysis that removes the confounding factor RT. Consequently, no statistically significant conclusion can be drawn from the authors' Bayesian model comparison analysis. Overall, the analyses reported in the supplementary information appears to rule out the claims the authors made about motor activations and shows that motor activations are simply related to motor responses, as expected from previous studies. Thus, the authors need to remove from their manuscript including the abstract, all the statements related to motor activations, in particular those in the Discussion stating that the motor cortex may integrate values (possibly encoded in the vmPFC) and other decision-related variables (note also that problematically, there are no connections between the vmPFC and the motor cortex). Additionally, the main manuscript should explicitly mention that when controlling for RTs, motor activations become unrelated to DV.

In conclusion, I think the revised paper presents a valid replication of Badre et al. (2012) using a

behavioral protocol with some advantages and disadvantages, but provides no new insights supported by the data.

Reviewer #3 (Remarks to the Author):

I appreciate the authors' efforts to respond to my previous reviews and the details they provide regarding the initial bug in their analyses. The present study uses a behavioral protocol dissociating total and relative uncertainty, while in Badre et al. (2012) these variables are orthogonalized in the regression analyses. The present study reports activations replicating those previously reported in Badre et al. (2012). The authors then claim in the paper that their study provides new insights as the results are derived from a normative adaptive model predicting activations associated with V/TU and reveal motor activations associated with the decision variable. However, the revised paper along with the additional analyses regarding motor activations still provides clear evidence against the authors' claims:

A- A first major problem is that in contradiction with the prediction, they found no activations associated with V/TU. The authors then explain in the discussion that this variable might not be explicitly encoded in neural networks implementing sequential sampling models. If the authors believe so and if they want their paper to go beyond Badre et al. (2012) and beyond a "wishful thinking", as they claim, they have to carry out the analyses comprising sequential sampling regressors rather than V/TU

To support our interpretation in terms of sampling, we performed an analysis showing that the residual variance in the decision value signal is correlated with (squared) total uncertainty, as predicted by a sampling mechanism. We added the following to the results section:

"Variability in the decision value signal scales with total uncertainty

The lack of any main effect for V/TU in GLM 1 could be explained by a mechanistic account according to which, instead of directly implementing our closed-form probit model (Eq. 4), the brain is drawing and comparing samples from the posterior value distributions. This corresponds exactly to Thompson sampling and would produce the exact same behavior as the analytical model. However, it makes different neural predictions, namely that: 1) there would be no explicit coding of V/TU, and 2) the variance of the decision value would scale with (squared) total uncertainty. The latter is true because the variance of the Thompson sample for arm k on trial t is $\sigma_t^2(k)$, and hence the variance of the sample difference is $\sigma_t^2(1) + \sigma_t^2(2) = TU^2$. Thus while we cannot infer the drawn samples on any particular trial, we can check whether the unexplained variance around the mean decision value signal in left M1 is correlated with TU^2 .

To test this hypothesis, we correlated the residual variance of the GLM 2 fits in the decision value ROI (Figure 5A; left M1, peak MNI [-38 -8 62]) with TU^2 . We found a positive correlation ($t(30) = 2.06$, $p = 0.05$, two-tailed t-test across subjects of the within-subject Fisher z-transformed Pearson correlation coefficients), consistent with the idea that total uncertainty affects choices via a sampling mechanism that is implemented in motor cortex.”

And the following to the results section:

“Residual Variance Analysis

To show that the variance of the decision value signal scales with total uncertainty, we extracted the residuals of the GLM 2 fits from the DV ROI (Figure 5; left M1, peak MNI [-38 -8 62]), averaged within a 10-mm sphere around the peak voxel. As with the decoding analysis, we accounted for the hemodynamic lag by taking the residuals 5 s after trial onset to correspond to the residual activations on the given trial. We then performed a Pearson correlation between the square of the residuals (the residual variance) and TU^2 across trials for each subject. Finally, to aggregate across subjects, we Fisher z-transformed the resulting correlation coefficients and performed a two-tailed one sample t-test against zero.”

We also added the following to the abstract:

“The variance of this decision value signal scaled with total uncertainty, consistent with a sampling mechanism for random exploration.”

And the following to the discussion:

“Such a sampling mechanism also predicts that the variance of the decision value signal should scale with (squared) total uncertainty, which is precisely what we found. Overall, our data suggest that random exploration might be implemented by a sampling mechanism which directly enters the drawn samples into the decision value computation in motor cortex.”

B- A second major, even more critical problem is that the additional analyses reported in the revised supplementary information show the evidence that in contradiction with authors' claims, motor activations are actually unrelated to the Decision Variable (DV) but associated with Reaction Times (RT). When controlling for the confounding factor RT, as I requested in my previous review and as systematically performed in previous studies investigating neural correlates of DVs, the authors found that in the regression analysis, motor

activations were unassociated with variable DV. The conclusion should be that consistent with the literature, motor activations reflect motor responses instead of decision variable DV. However, to overcome this problem, the authors carried out a notoriously, statistically biased analysis. The analysis falls into the double-dipping fallacy (Kriegeskorte et al., Nature Neuroscience, 2009) as performed on data selected within the cluster associated with DV according to the regression analysis that removes the confounding factor RT. Consequently, no statistically significant conclusion can be drawn from the authors' Bayesian model comparison analysis

We circumvented the circularity issue using cross-validation. Now the data we use for ROI selection model comparison are completely independent. We added the following to the supplement:

“Specifically, following our previous work (Tomov et al., 2019), we approximated the log model evidence as $-0.5 * BIC$, where the BIC was computed based on the residual variance of the GLM fits within a 10 mm sphere around the peak voxel in left M1 from GLM 2. To prevent circularity (Kriegeskorte et al., 2009), we performed this using leave-one-subject-out cross-validation: for each subject, we computed the BIC in the peak ROI from the group-level DV contrast computed using all other subjects. Since SPM fits each subject separately, this means that we used independent data for ROI selection and model comparison, resulting in an unbiased analysis. To ensure the validity of our inference, we confirmed that the resulting ROIs were highly overlapping (Figure S10), with all but one subject having the same left M1 ROI as the contrast using all subjects (Figure 5A, MNI [-38 -8 62]). This analysis strongly favored GLM 2A (PXP = 0.96) over GLM 2 (PXP = 0) and GLM 2B (PXP = 0.04)”

Overall, the analyses reported in the supplementary information appears to rule out the claims the authors made about motor activations and shows that motor activations are simply related to motor responses, as expected from previous studies. Thus, the authors need to remove from their manuscript including the abstract, all the statements related to motor activations,

We hope that our new analyses make the case that motor cortex indeed computes a decision value via a sampling mechanism, so we decided to keep the statements in the abstract and discussion.

in particular those in the Discussion stating that the motor cortex may integrate values (possibly encoded in the vmPFC) and other decision-related variables (note also that problematically, there are no connections between the vmPFC and the motor cortex).

We removed the suggestion that motor cortex is integrating values from vmPFC.

Additionally, the main manuscript should explicitly mention that when controlling for RTs, motor activations become unrelated to DV.

We added the following to the results:

“Another possible confound is reaction time (RT). When controlling for RT, motor cortex activations become unrelated to DV (GLM 2A in Supplemental Information). This could be explained by a sequential sampling implementation of our model (see Discussion), according to which RT would depend strongly on DV. Consistent with this interpretation, model comparison revealed that left M1 activity is best explained by a combination of DV and RT, rather than DV or RT alone (see Supplemental Information).”

Additionally, we would like to point out that we find the practice of controlling for RT's by including a RT regressor to be not that common, precisely because of the strong relationship between DV and RT (e.g. De Martino, 2013, NatNeurosci).

In conclusion, I think the revised paper presents a valid replication of Badre et al. (2012) using a behavioral protocol with some advantages and disadvantages, but provides no new insights supported by the data

We hope our additional analyses address the concerns of the reviewer. Regarding the novelty of our results, we respectfully beg to differ. Even without considering the sampling result and motor cortex result, we believe that our results go beyond the Badre et al. study, which 1) did not orthogonalize RU and TU in the task design, 2) was underpowered by modern standards (only 15 subjects), 3) did not provide a computational rationale for why TU would be computed by the brain at all, and 4) did not show that variance in the TU encoding could predict variance in behavior. Our results bring the previous result of Badre et al. under a broad normative framework for uncertainty-guided exploration and show how this framework maps onto different brain regions.

****REVIEWERS' COMMENTS:**

Reviewer #3 (Remarks to the Author):

The authors properly responded to my comments. I think their last revisions and the inclusion of sampling analyses improved a lot the paper and make the results meaningful and beyond previous work from Badre et al.. In my opinion, the paper is now suitable for publication

As a final advice to the authors: while they said they cannot infer the drawn samples on any particular trial, advanced particle filtering methods actually exist to make such analyses, which in particular allow to marginalising out over sampling distributions and to properly compute model posterior probabilities. But this approach certainly goes beyond the scope of the present study.

Reviewer #3 (Remarks to the Author):

The authors properly responded to my comments. I think their last revisions and the inclusion of sampling analyses improved a lot the paper and make the results meaningful and beyond previous work from Badre et al.. In my opinion, the paper is now suitable for publication

As a final advice to the authors: while they said they cannot infer the drawn samples on any particular trial, advanced particle filtering methods actually exist to make such analyses, which in particular allow to marginalising out over sampling distributions and to properly compute model posterior probabilities. But this approach certainly goes beyond the scope of the present study.

We thank the reviewer for the suggestion and we agree that it is beyond the scope of the present work.